# Adaptive Online Replanning with Diffusion Models

**Siyuan Zhou[1], Yilun Du[2], Shun Zhang[3], Mengdi Xu[4], Yikang Shen[3],**
**Wei Xiao[2], Dit-Yan Yeung[1], Chuang Gan[3,5]**

[1]Hong Kong University of Science and Technology
[2]Massachusetts Institute of Technology, [3]MIT-IBM Watson AI Lab
[4]Carnegie Mellon University, [5]UMass Amherst

## Abstract

Diffusion models have risen as a promising approach to data-driven planning, and have demonstrated impressive robotic control, reinforcement learning, and video planning performance. Given an effective planner, an important question to consider is replanning – when given plans should be regenerated due to both action execution error and external environment changes. Direct plan execution, without replanning, is problematic as errors from individual actions rapidly accumulate and environments are partially observable and stochastic. Simultaneously, replanning at each timestep incurs a substantial computational cost, and may prevent successful task execution, as different generated plans prevent consistent progress to any particular goal. In this paper, we explore how we may effectively replan with diffusion models. We propose a principled approach to determine when to replan, based on the diffusion model's estimated likelihood of existing generated plans. We further present an approach to replan existing trajectories to ensure that new plans follow the same goal state as the original trajectory, which may efficiently bootstrap off previously generated plans. We illustrate how a combination of our proposed additions significantly improves the performance of diffusion planners leading to 38% gains over past diffusion planning approaches on Maze2D, and further enables the handling of stochastic and long-horizon robotic control tasks. Videos can be found on the anonymous website: https://vis-www.cs.umass.edu/replandiffuser/.

## 1 Introduction

Planning is a pivotal component in effective decision-making, facilitating tasks such as collision avoidance in autonomous vehicles [29], efficient and long-horizon object manipulation [12] and strategic reasoning for extended periods [32, 25]. Recently, diffusion has risen as a promising approach to data-driven planning [17, 1, 14, 40], enabling impressive advancements in robot control [6] and the generation of video-level plans [9].

In many planning settings, an important variable to consider is not only *how to plan*, but also *when it is time to replan*. Environments are often stochastic in nature, making previously constructed plans un-executable given an unexpected environmental change. Moreover, inaccurate execution of individual actions in an environment will often also cause states to deviate from those in the expected plan, further necessitating replanning. Finally, environments will often be partially observable, and the optimal action or sequence of actions may deviate substantially from those originally planned, especially when new information comes to light.

Previous works in planning with diffusion models have not explored how to tackle these factors, and typically generate new plans at each iteration. Such a process is computationally expensive as a

37th Conference on Neural Information Processing Systems (NeurIPS 2023).

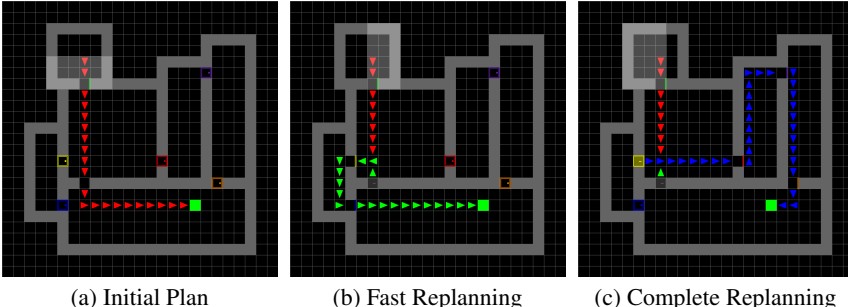

| (a) Initial Plan | (b) Fast Replanning | (c) Complete Replanning |

Figure 1: **Online planning procedure. (a)** The agent first generates a plan from scratch at the start of the task. **(b)** The agent finds the first door inaccessible and replans a new trajectory that deviates slightly from the original plan. **(c)** The agent finds the yellow door also locked, and replans a completely different trajectory from scratch.

generated plan in diffusion models requires hundreds of sampling steps. Frequent replanning may also be detrimental to task completion, as rapid switching between plans impedes persistent progress towards a single final goal [18]. In this paper, we propose a principled approach to adaptively online **R**eplanning with **D**iffusion **M**odels, called RDM. In RDM, we propose to leverage the internal estimated likelihood of the currently executed plan as a gauge on when is a good time to replan. At the start of plan execution, a predicted plan is freshly sampled from the diffusion model and has a high estimated likelihood. As the plan gets executed, or when unexpected state changes happen, the underlying transitions expected in the plan get increasingly less likely, and the internally estimated likelihood of the diffusion model tells us to replan.

Given a previously generated plan, how do we effectively regenerate a new plan? Directly resampling the entire plan can be computationally expensive, as all the previous computations put in the previous plan would be wasted. Furthermore, a newly generated plan might erase all the progress we made previously toward the goal from a previous plan. Instead, based on the feasibility of the current plan being executed, which is estimated by likelihood determined by the diffusion model, we propose to partially replan based on our plan, or fully replan from scratch. We initialize our new plan with the old one and add a small amount of noise to the plan before then denoising the added noise. This light amount of denoising serves as a planning refinement operator.

We illustrate our online planning procedure in Figure 1, where a learned agent first generates a full plan for the goal as in Figure 1a. When the agent reaches the first door, the likelihood of the plan drops, as the agent finds it locked, and the corresponding solution becomes infeasible due to the door's inaccessibility. However, the agent can slightly adjust the initial plan by instead entering the left yellow door and taking a short detour, after which the rest of the plan remains the same. Based on the estimated likelihood of the plan, we may replan and regenerate a new solution based on the previous solution as shown in Figure 1b. In another situation where the yellow door is also locked, and there is no nearby solution, our likelihood plummets, telling us that we need to regenerate a new and completely different solution from scratch as shown in Figure 1c.

In summary, our contributions are three-fold. **(1)** We introduce RDM, a principled approach to determining when to replan with diffusion models. **(2)** We illustrate how to effectively replan, using a previously generated plan as a reference. **(3)** We empirically validate that RDM obtains strong empirical performance across a set of different benchmarks, providing a 38% boost over past diffusion planning approaches on Maze2D and further enabling effective handling of stochastic and long-horizon robotic control tasks.

## 2 Background

### 2.1 Problem Setting

We consider a trajectory optimization problem similar to Diffuser [17]. A trajectory is represented as $\tau = (s_0, a_0, s_1, a_1, \ldots, s_T, a_T)$, where $s_t, a_t$ are the state and action at the $t$-th time step, and $T$ is the episode length. The dynamics function of the environment is formulated as $s_{t+1} = f_\sigma(s_t, a_t)$, where $\sigma$ represents the stochastic dynamics. For domains with partial observations, the agent observes an observation $o_t$ instead of the state $s_t$. The objective of the trajectory optimization problem is to find a sequence of actions $a_{0:T}$ that maximizes $\mathcal{J}(s_0, a_{0:T})$, which is the expected sum of rewards

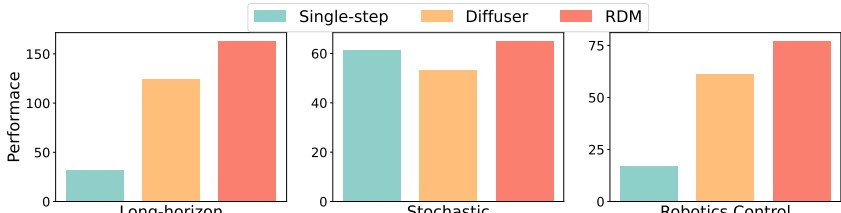

Figure 2: **Results overview.** The performances of RDM significantly outperform the models with single-step execution (i.e. IQL) and previous planning-based models with diffusion models (i.e. Decision Diffuser) across long-horizon planning (Maze2D), stochastic environments (D4RL locomotion) and robotic control (RLBench). The performance metrics are normalized returns for D4RL tasks (including Maze2D and Locomotion) and success rates for RLBench.

over all the time steps,

$$\arg\max_{a_{0:T}} \mathcal{J}(s_0, a_{0:T}) = \arg\max_{a_{0:T}} \mathrm{E}_{s_{1:T}} \sum_{t=0}^{T} r(s_t, a_t).$$

Following the Decision Diffuser [1], we plan across state-space trajectories and infer actions through inverse dynamics.

## 2.2 Planning with Diffusion Models

In Diffuser [17], plan generation is formulated as trajectory generation $\tau$ through a learned iterative denoising diffusion process $p_\theta(\tau^{i-1} \mid \tau^i)$ [33]. This learned denoising process is trained to reverse a forward diffusion process $q(\tau^i \mid \tau^{i-1})$ that slowly corrupts the structure of trajectories by adding noise. Both processes are modeled as Markov transition probabilities:

$$q(\tau^{0:N}) = q(\tau^0) \prod_{i=1}^{N} q(\tau^i|\tau^{i-1}), \quad p_\theta(\tau^{0:N}) = p(\tau^N) \prod_{i=1}^{N} p_\theta(\tau^{i-1}|\tau^i). \tag{1}$$

Given a set of predetermined increasing noise coefficients $\beta_1, ..., \beta_N$, the data distribution induced is given by:

$$q(\tau^i \mid \tau^{i-1}) := \mathcal{N}(\sqrt{1 - \beta_i}\tau^{i-1}, \beta_i I), \quad p_\theta(\tau^{i-1} \mid \tau^i) := \mathcal{N}(\mu_\theta(\tau^i, i), \sigma_i^2 I), \tag{2}$$

where $q(\tau^0)$ is the data distribution and $p(\tau^N)$ is a standard Gaussian prior.

Diffuser is trained to maximize the likelihood of trajectories through a variational lower bound over the individual steps of denoising

$$\mathrm{E}_{q(\tau^0)}[\log p_\theta(\tau^0)] \geq \mathrm{E}_{q(\tau^{0:N})}\left[\log \frac{p_\theta(\tau^{0:N})}{q(\tau^{1:N}|\tau^0)}\right] \approx \mathrm{E}_{\tau^{1:N} \sim q(\tau^{1:N})}\left[\sum_{i=1}^{N} \log p_\theta(\tau^{i-1}|\tau^i)\right], \tag{3}$$

where trajectory plans may then be generated by sampling iteratively from the learned denoising network for $N$ steps. We follow the formulation of planning in Decision Diffuser [1], where if we seek to optimize an objective $\mathcal{J}(\tau)$, we directly learn and sampling from conditional trajectory density $p_\theta(\tau|\mathcal{J})$.

In terms of notations, we follow Janner et al. [17] and use superscripts $i \in [0, N]$ to denote diffusion steps (for example, $\tau^i$), and use subscripts $t \in [0, T]$ to denote time steps of the agent's execution in the environment (for example, $s_t, a_t$, etc.).

## 3 Adaptive Online Replanning with Diffusion Models

In this section, we present RDM, an approach to online replanning with diffusion models. First, we discuss different criteria for determining *when* a solution is problematic and needs replanning. We then introduce the estimated likelihood of a planned trajectory as a criterion to determine when to replan in Section 3.1. In Section 3.2, we discuss *how* we replan by comparing three distinct replanning strategies and propose an adaptive replanning approach, based on combinations of the two approaches.

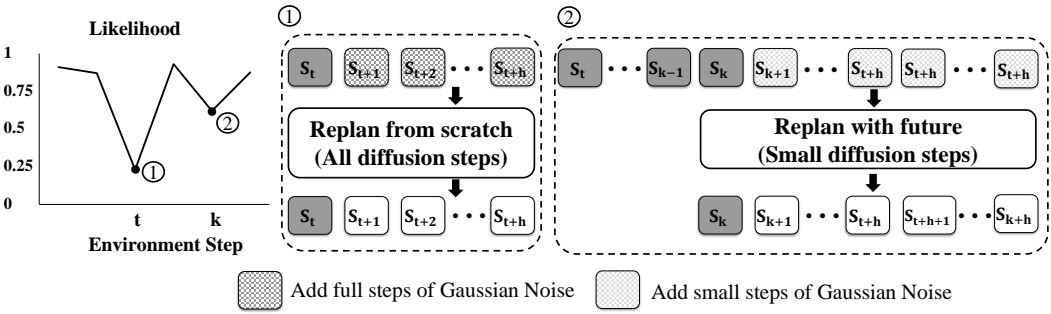

Figure 3: **Overview of our replanning approach RDM.** The first figure shows the likelihood curve of the sampled trajectory as the environmental steps increase. There are two steps $t$ and $k$ with low probabilities. Step $t$ corresponds to Replan from scratch in the middle figure, where RDM regenerates a completely new trajectory based only on the current state and from Gaussian noise. Step $k$ corresponds to Replan with future in the right-most figure, where RDM shifts the current state as the first state and repeats the last state to fill the rest. RDM also introduces some timesteps of noise to the trajectory and denoises.

## 3.1 When to Replan

When the agent sequentially executes a plan $\tau = (s_0, a_0, s_1, a_1, \ldots, s_T, a_T)$, different factors in the environment such as unexpected interventions or imperfect transitions may cause future states and actions in the plan to deviate substantially from the desired trajectory. We denote with $\tau_{\rightarrow k}$ the trajectory executed in the environment up to time step $k$. Formally, $\tau_{\rightarrow k} = (s_0, a_0, \bar{s}_1, \bar{a}_1, \ldots, \bar{s}_k, \bar{a}_k, s_{k+1}, \ldots, s_T, a_T)$, where $s_t, a_t$ are planned states and actions generated by the diffusion model; $\bar{s}_t$ is a state observed in the environment, and $\bar{a}_t$ is an action that is computed using the inverse dynamics during the execution process (which is the action transiting to $s_{t+1}$ from $\bar{s}_t$). Continuing to execute the current plan will result in more significant discrepancies and potentially catastrophic failures. It may not even be feasible to transit to the next planned state ($s_{k+1}$) from the current state that the agent is at ($\bar{s}_k$). In such cases, we need to update the planned trajectory based on $\tau_{\rightarrow k}$, a process we refer to as *replanning*.

Unfortunately, determining the need for replanning is non-trivial. Directly replanning at a fixed interval is straightforward but inflexible, and may come too late when the environment suddenly changes. It can also make the planning algorithm computationally inefficient when the planned trajectory is executed perfectly and does not require replanning. Instead, it is essential to establish a criterion that automatically determines when replanning is necessary. An alternative approach is to replan when the distance between the actual state observed from the environment and the desired state in the plan exceeds a certain threshold. However, in many cases, even if the actual state the agent is in matches with the state in the plan, new information about the environment may make the current plan infeasible. Thus, it is desirable to come up with a comprehensive and adaptive approach to determine the appropriate timing for replanning.

In this paper, we propose a novel method that utilizes the inherent estimated likelihood of the trajectory in diffusion models as a criterion for replanning. As the diffusion model is trained on different successful trajectories in an environment, its learned likelihood function implicitly captures the executability of an underlying plan. A feasible trajectory will be assigned a high probability, with increasingly less feasible trajectories assigned decreasing probabilities. Thus, when the likelihood assessment of a diffusion model on the plan has a low probability, it is a good indicator of the need for replanning.

The key challenge is to estimate the likelihood of a partially-executed trajectory $\tau_{\rightarrow k}$. Recall that we optimize the variational lower bound defined in Equation 3. We propose to use the same objective to determine the likelihood of a partially-executed trajectory. Intuitively, we assess *the degree to which a diffusion model recovers its original plan after the plan has been corrupted by Gaussian noise*. Concretely, given $\tau_{\rightarrow k}$, we added different magnitudes of Gaussian noise to the trajectory, corresponding to sampling based on the diffusion process $q(\tau^i \mid \tau_{\rightarrow k})$ at a diffusion step $i$. We then compute the KL divergence between the posterior $q(\tau^{i-1} \mid \tau^i, \tau_{\rightarrow k})$ and the denoising step $p_\theta(\tau^{i-1} \mid \tau^i)$ at the diffusion step $i$. We estimate likelihood by computing this KL divergence at several fixed different diffusion steps $i$, which we discuss in detail in the appendix.

---

**Algorithm 1 When to Replan.**

---

1: **Input:** Diffusion model $p_\theta$, Planning horizon $H$, Episode length $T$, Threshold $l_s$ of **Replanning from scratch**, Threshold $l_f$ of **Replanning with future** and $l_s < l_f$, Diffusion step $N_s$ of **Replan from scratch**, Diffusion step $N_f$ of **Replan with future** and $N_f \ll N_s$, Diffusion steps $I$ for computing likelihood
2: Environment step $t = 0$
3: Generate a trajectory $\tau_t = (s_t, a_t, \ldots, s_{t+H}, a_{t+H})$
4: **for** $t$ in $0, \ldots, T$ **do**
5:     Extract $(s_t, s_{t+1}, a_t)$ from $\tau_t$
6:     Execute $a_t$ and obtain $s_{t+1}$ from the environment
7:     Replace the $s_{t+1}$ in $\tau_t$
8:     $\tau^0 \leftarrow \tau_t$
9:     Sample $\tau^i \sim q(\tau^i \mid \tau^0)$ for all $i \in I$
10:     Compute average likelihood $L_t = \frac{1}{|I|} \sum_{i \in I} D_{KL}(q(\tau^{i-1} \mid \tau^i, \tau^0) \| p_\theta(\tau^{i-1} \mid \tau^i))$
11:     **if** $(t+1)\%H = 0$ or $L_t <= l_1$ **then**
12:         Sample a trajectory $\tau_t$ with **Replanning from scratch**
13:     **else if** $L_t <= l_2$ **then**
14:         Sample a new trajectory $\tau_t$ based on previous trajectory $\tau_{t-1}$ with **Replanning with future**
15:     **else**
16:         $\tau_{t+1} \leftarrow \tau_t$
17:     **end if**
18: **end for**

---

**Algorithm 2 Replanning from scratch**

---

1: **Input:** Diffusion model $p_\theta$, Diffusion steps $N_s$, Conditions $C$
2: Initialize plan $\tau^{N_s} \sim \mathcal{N}(\mathbf{0}, \boldsymbol{I})$
3: **for** $i$ in $N_s, \ldots, 1$ **do**
4:     $\tau^{i-1} \leftarrow p_\theta(\tau^i)$
5:     Constraint $\tau^{i-1}$ with conditions $C$
6: **end for**
7: **Return:** $\tau^0$

---

**Algorithm 3 Replanning with future**

---

1: **Input:** Diffusion model $p_\theta$, Diffusion process $q$, Diffusion steps $N_f$, Conditions $C$, Previous sampled trajectory $\tilde{\tau}$
2: $\tau^{N_f} \leftarrow q_{N_f}(\tilde{\tau})$
3: **for** $i$ in $N_f, \ldots, 1$ **do**
4:     $\tau^{i-1} \leftarrow p_\theta(\tau^i)$
5:     Constraint $\tau^{i-1}$ with conditions $C$
6: **end for**
7: **Return:** $\tau^0$

---

Given this likelihood estimate, we set a threshold for the likelihood of the planning trajectories. If the likelihood drops below the threshold, it indicates the current plan deviates significantly from the optimal plan and can cause the failure of the plan, which suggests that replanning is necessary. We illustrate this replanning process in Algorithm 1.

## 3.2 How to Replan

Given a previous partially executed plan executed to time step $k$, $\tau_{\rightarrow k}$, and suppose we decide to replan at this current time step using one of the criteria defined the Section 3.1, how do we construct a new plan $\tau'$? We consider three separate replanning strategies, which are the following:

1. **Replan from scratch.** In this strategy, we regenerate a completely new planning trajectory based only on the current state. This involves completely resampling the trajectory starting from Gaussian noise and running all steps (defined as $N_s$) of iterative denoising. Although such an approach fully takes advantage of the diffusion model to generate the most likely trajectory, it is computationally expensive and discards all information in the previously generated plan (Algorithm 2).

2. **Replan from previous context.** Instead, we may replan by considering both the current and historical states when executing the previously sampled plan. Given a partially-executed trajectory $\tau_{\rightarrow k}$ and current time step $k$, we sample a new trajectory $\tau'$ based on $\tau_{\rightarrow k}$. Specifically, we run $N_p$ diffusion steps to add noise to $\tau_{\rightarrow k}$ at all the timesteps that are larger than $k$ by sampling from the forward process $q(\tau^{N_p} \mid \tau_{\rightarrow k})$. We then run $N_p$ steps of denoising. Here, $N_p$ is much smaller than Replan from scratch. We will fix the timesteps of $\tau_{\rightarrow k}$ that are smaller than $k$ to their observed

value, as we have already executed them in the environment. This approach can be seen as locally repairing a predicted plan, and thus requires a substantially smaller number of diffusion steps.

3. **Replan with future context.** In a similar way to replan with previous context, in replan with future context, we initialize our new plan $\tau'$ based on our partially-executed plan $\tau_{\to k}$. However, in contrast to the previous approach, we do not add the previously executed states of the plan. We remove the executed states in $\tau_{\to k}$ and repeatedly add the last predicted state in $\tau_{\to k}$ on the right to fill in the remaining plan context. Namely, $\tilde{\tau} = (\bar{s}_k, \bar{a}_k, s_{k+1}, a_{k+1}, \ldots, s_T, a_T, s_T, a_T, \ldots, s_T, a_T)$, where $s_T, a_T$ are repeated to match the input length of the diffusion model. To replan, similar to replan from previous context, we add $N_f$ steps of noise to $\tilde{\tau}$ by sampling from the forward process $q(\tau^{N_f} \mid \tau_{\to k})$ and then denoise for $N_f$ steps (Algorithm 3), where $N_f$ is much smaller than $N_s$.

Replanning from scratch is highly time-consuming and inefficient for real-time tasks since it requires all diffusion steps to be run to generate a plan from Gaussian noise. However, in cases of significant interventions, it becomes necessary to regenerate the plan from scratch. Both replanning for future context or past context are efficient and locally repair the previously generated plan. Replanning with past context remains more consistent with the original goal of the previous plan, while replanning for future context further extends the current plan to future context.

We empirically find that replanning with future context is a more effective approach. In our proposed approach, based on the estimated likelihood of a current plan, we either replan from scratch or replan with future context. We illustrate the psuedocode for our approach in Algorithm 1. We provide a detailed ablation analysis of the different replanning strategies in the experiment section (Figure 7).

## 4 Experiments

In this section, we empirically use our proposed RDM algorithm for replanning in multiple decision-making tasks (as illustrated in Figure 2) and evaluate the performance of the generated trajectories. We empirically answer the following questions: **Q1:** In domains with long horizon plans, does RDM effectively and efficiently fix the initial plan? **Q2:** In domains with unexpected interventions, which makes the initial plan no longer feasible during the execution, does RDM efficiently find a feasible plan? **Q3:** How do when to replan and different replanning strategies affect the agent's performance? In all the experiments below, we report the results using five random seeds.

### 4.1 Long Horizon Planning

We first evaluate the long-horizon planning capacity of RDM and the baseline methods in the Maze2D environment. In this environment, the large Maze2D domain is especially challenging, as it may require hundreds of steps to reach the target and we found that baseline algorithms struggled to find perfect trajectories given the long horizon and sparse reward setting.

Maze2D is a goal-conditioned planning task, where given the starting location and the target location, the agent is expected to find a feasible trajectory that reaches the target from the starting location avoiding all the obstacles. The agent receives a reward of 1 when it reaches the target state and receives a reward of 0 otherwise. The starting and target locations are randomly generated, consistent with the setting in the Diffuser [17]. We evaluate replanning using our RDM algorithm and multiple competing baseline algorithms, including model-free reinforcement learning algorithms such as Batch-Constrained deep Q-learning (BCQ) [11], Constrained Q-learning (CQL) [22], Implicit Q-learning (IQL) [21], and planning-based algorithms like Diffuser [17].

We report the results in Table 1 where numbers for all baselines are taken from Diffuser [17]. As expected, model-free reinforcement learning algorithms (BCQ, CQL, and IQL) struggle to solve long-horizon planning tasks. On the other hand, Diffuser generates a full plan and outperforms these approaches, but does not replan online. As a result, it still does not perform well when the size of the domain increases. As the planning horizon increases, it is likely that some of the transitions generated by the diffusion model are infeasible as shown in Figure 4c. We find that our replanning approach, RDM, significantly outperforms all the baseline algorithms. Specifically, it achieves a score of 190% in the Maze2D Large setting, which is almost 63% higher than the Diffuser. We also visualize the sampled trajectory and the executed trajectories in Figure 4. We can observe that infeasible transitions and collisions with the wall will drop the likelihood substantially. RDM is able to find these infeasible transitions and replan accordingly.

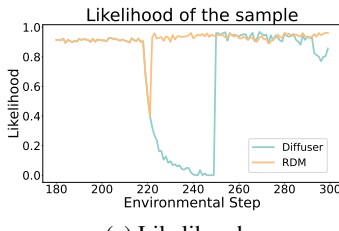 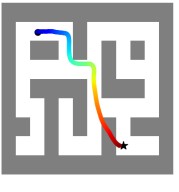 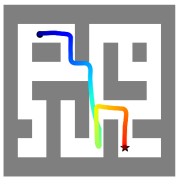 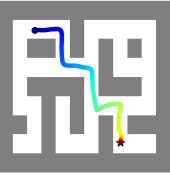

| (a) Likelihood | (b) Sample | (c) Diffuser | (d) RDM |

Figure 4: **Visualization on Maze2D.** **(a)** illustrates the likelihood curve of the sampled trajectory as the environmental steps increase, with a noticeable drop when the agent collides with the wall. **(b)** presents the sampled trajectory at the onset of the task. **(c)** demonstrates the actual trajectory of Diffuser, which follows a fixed interval replanning strategy. It is evident that the agent does not recognize the flawed part of the sampled trajectory, leading it to a less favorable state and resulting in wasted environmental steps. **(d)** shows the actual trajectory of RDM. In contrast to **(c)**, the agent successfully detects collisions and replans a feasible trajectory.

| Environment | | BCQ | CQL | IQL | Diffuser | RDM (Ours) |
|---|---|---|---|---|---|---|
| Maze2D | U-Maze | 12.8 | 5.7 | 47.4 | 113.9 | **133.6** ±3.0 |
| Maze2D | Medium | 8.3 | 5.0 | 34.9 | 121.5 | **159.3** ±3.1 |
| Maze2D | Large | 6.2 | 12.5 | 58.6 | 123.0 | **185.3** ±11.7 |
| **Average** | | 9.1 | 7.7 | 47.0 | 119.5 | **159.4** |
| Multi2D | U-Maze | - | - | 24.8 | 128.9 | **152.1** ±2.31 |
| Multi2D | Medium | - | - | 12.1 | 127.2 | **155.1** ±4.0 |
| Multi2D | Large | - | - | 13.9 | 132.1 | **195.4** ±7.6 |
| **Average** | | - | - | 16.9 | 129.4 | **167.5** |

Table 1: **Long horizon planning.** The normalized returns of RDM and other baselines on Maze2D. RDM achieves 38% gains beyond Diffuser for all tasks and even 63% gains for large maps.

## 4.2 Stochastic Environments

Next, we investigate whether RDM can tackle environments with stochastic dynamics. We consider a stochastic variation of the D4RL locomotion benchmark [10] by adding randomness to the transition function. Stochasticity can make the trajectory optimization problem more challenging, as the executed trajectory can be significantly different from the initially planned trajectory. We omitted the `Halfcheetah` environment as we found all algorithms performed poorly with added stochasticity. In this setting, we consider competing baseline methods including a state-of-the-art model-free algorithm, IQL [21], a Transformer-based RL algorithm, Decision Transformer (DT) [5], and a diffusion model-based algorithm, Decision Diffuser (DD) [1].

As shown in Table 2, although the performance of all the methods drops significantly compared with one without stochasticity, RDM performs comparatively well among all baselines. RDM with replanning strategy is able to detect such stochasticity and quickly regenerate a new plan, resulting in improved performance. We compare DD and RDM using the same number of total diffusion resampling steps.

## 4.3 Robotic Control

We further evaluate the RDM's capacity in complex domains with visual observations. We consider the RLBench domain [15], which consists of 74 challenging vision-based robotic learning tasks. The agent's observations are RGB images captured by multi-view cameras. The agent controls a 7 DoF robotic arm. The domain is designed to reflect a real-world-like environment with high dimensionality in both the observation space and the action space, making it challenging to solve (Figure 5b).

Most previous works tackle the RLBench tasks by taking actions in macro-steps [13], where the agent switches to a different macro-step when the state of the gripper or the velocity changes. However, such an approach has several limitations: **(1)** The longest trajectory length achievable using macro steps is relatively short, which is slightly over 30 steps. Such a limitation hinders the agent's ability

| Dataset | Environment | IQL | DT | DD | RDM (Ours) |
|---------|-------------|-----|-----|-----|------------|
| **Med-Expert** | **Hopper** | 48.9 | 52.5 | 49.4 | **59.4** ±3.4 |
| **Med-Expert** | **Walker2d** | **90.7** | 90.4 | 71.1 | **92.5** ±2.2 |
| **Medium** | **Hopper** | 47.6 | 49.9 | 45.3 | **53.2** ±1.2 |
| **Medium** | **Walker2d** | **71.8** | 67.9 | 60.7 | **70.7** ±3.0 |
| **Med-Replay** | **Hopper** | **55.3** | 40.8 | 47.2 | **57.9** ±3.2 |
| **Med-Replay** | **Walker2d** | 54.4 | 52.6 | 46.4 | **57.3** ±3.1 |
| **Average** | | 61.5 | 59.0 | 53.4 | **65.2** |

Table 2: **Stochastic environments**. The normalized returns of RDM and other baselines on Locomotion with randomness. RDM exhibits comparable or superior performance.

**Fail to open box**   **Replan and retry**

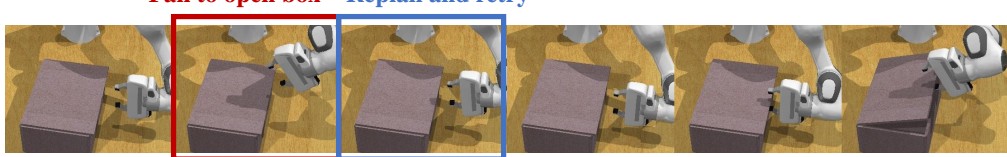

(a) With our replanning approach

**Fail to open box**   **Continue the plan**

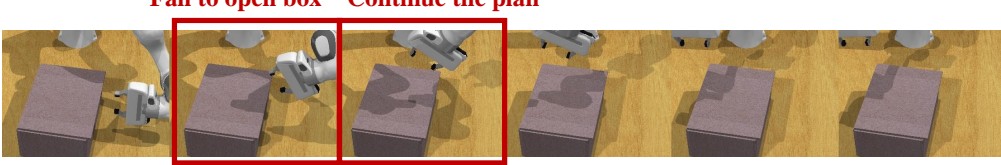

(b) Without our replanning approach

Figure 5: **Comparison between trajectories with and without replanning.** **(a)** Our replanning approach allows the agent to detect when the robotic arm fails to open the box and instructs it to retry. **(b)** Without our replanning approach, the failure goes unnoticed, leading the arm to persist in following the previous, unsuccessful plan.

to solve long-horizon tasks. **(2)** Macro-steps are pre-defined, which usually require expert knowledge or demonstrations. **(3)** Macro-steps constrain the agent's action space, which makes it impossible to optimize at a finer granularity (for example, avoiding an intermediate collision) and prevent the agent from doing fine-grained control. Therefore, we do not use macro-step in our experiments.

We choose the following three tasks for evaluation: `open box`, `close box`, and `close fridge`. The experimental results are in Table 3. We find that baseline algorithms fail to accomplish these three tasks. In contrast, RDM outperforms all the baselines by a large margin and achieves a success rate of 77.3%. We visualize the execution of RDM in Figure 5. RDM successfully detects failures in the initial plan and efficiently replans during the execution.

| Environment | BC | IQL | DT | DD | RDM (Ours) |
|-------------|-----|-----|-----|-----|------------|
| `open box` | 9.3 | 17.6 | 9.8 | 52.9 | **72.0** ±7.5 |
| `close box` | 5.9 | 10.3 | 7.8 | 46.0 | **61.3** ±7.7 |
| `close fridge` | 13.0 | 23.1 | 21.7 | 84.6 | **98.6** ±2.7 |
| **Average** | 9.4 | 17.0 | 13.1 | 61.2 | **77.3** |

Table 3: **Robotic control.** The success rate of RDM and other baselines for three tasks: `open box`, `close box` and `close fridge` on RLBench. RDM outperforms baselines by a large margin.

## 4.4 Ablative Study

**Different Levels of Stochasticity**. We investigate how the algorithms respond to different levels of stochasticity by setting different noise values $\epsilon$ in the task `Hopper-medium-expert`, where at each time step, the transition is the same as the deterministic version of the domain with probability $1 - \epsilon$,

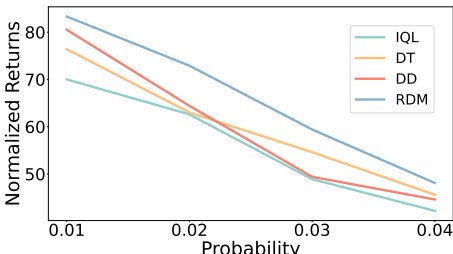

Figure 6: **Different levels of stochasticity.** The normalized returns with different probabilities $\epsilon$ of randomness in the task `Hopper-medium-expert`. By adaptive planning, RDM is more effective than other baselines.

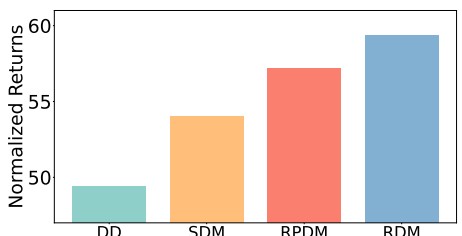

Figure 7: **Replan analysis**. We analyze replanning at fixed intervals (DD), replanning based on state distance (SDM), and replanning based on past context (RPDM). With the same number of diffusion steps on `Hopper-medium-expert`, RDM obtains the best performance.

and is random (as if a random action is taken) with probability $\epsilon$. The results are in Figure 6. We find that RDM consistently outperforms other baselines under all $\epsilon$ values.

**Different Replanning Strategies**. To analyze the different replan approaches, we consider the following baselines. (1) Decision Diffuser (DD) replans using two fixed intervals, for deciding when to replan from scratch and when to plan with future context, respectively. (2) Replan on State Distance (SDM) replans when the error between the observed states ($\bar{s}_t$) and desired states in the plan ($s_t$) exceeds a threshold. It replans with the future context. The following two algorithms determine when to plan according to the diffusion model likelihood. (3) Replan from previous context (RPDM) replans with on past context, and (4) RDM replans with future context. In Figure 7, we find that SDM and RPDM both have a better performance than DD, among which RDM achieves the best performance. All methods use the same number of total diffusion steps.

We provide additional ablation studies and visualization analysis in the Appendix.

## 5 Related Work

**Diffusion Models for Decision Making**. The use of diffusion models for decision-making tasks has seen increased interest in recent years [17, 1, 9, 6, 30, 40, 23, 14, 38, 24] and have been applied to tasks such as planning [17, 1], robotic control [6], behavior cloning [30], and video generation [9]. In [17, 1] introduce an approach to planning with diffusion models, but both methods typically replan at each iteration. In this paper, we propose a method to adaptively determine when to replan with diffusion models and illustrate different approaches to replan.

**Model-Based Planning**. Our work is also related to existing work in model-based planning [35, 4, 7, 19, 3, 36, 2, 27, 34, 8, 37, 28]. For domains with finite states, traditional methods use dynamic programming to find an optimal plan, or use Dyna-algorithms to learn the model first and then generate a plan [35]. For more complex domains, a learned dynamics model be autoregressively rolled out [7, 19], though gradient planning through these models often lead to adversarial examples [3, 36, 2]. To avoid this issue approaches have tried gradient-free planning methods such as random shooting [27], beam search [34], and MCMC sampling [4]. Our work builds on a recent trend in model-based planning which folds the planning process into the generative model [16].

## 6 Limitations and Conclusion

**Conclusion**. In this paper, we propose RDM, an adaptive online replanning method, with a principled approach to determine when to replan and an adaptive combined strategy for how to replan. We demonstrate RDM's remarkable proficiency in managing long-horizon planning, unpredictable dynamics, and intricate robotic control tasks. An interesting line of future work involves investigating the scalability of RDM to more complex and larger-scale problems and extending to real robot tasks and autonomous driving.

**Limitations**. There are several limitations to our approach. Firstly, the effectiveness of planning and replanning is heavily dependent on the generative capabilities of the diffusion models. If the diffusion

model is unable to accurately represent the current task, its plans may not be viable and RDM may fail to identify problematic trajectories. Secondly, despite RDM's significant improvement on the performance of previous diffusion planning methods, we observed that likelihood-based replanning was insensitive to some planning failures. This highlights the potential benefit of combining likelihood-based replanning with other indicators for replanning.

## Acknowledgment

This work has been made possible by a Research Impact Fund project (R6003-21) funded by the Research Grants Council of the Hong Kong Government.

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

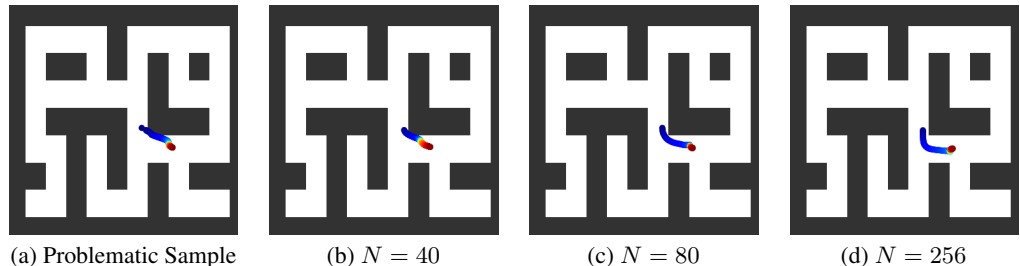

| (a) Problematic Sample | (b) $N = 40$ | (c) $N = 80$ | (d) $N = 256$ |

Figure 8: **Samples of replanning with different diffusion steps. (a)** demonstrates the problematic trajectory. **(b)** presents the sampled trajectory after replanning with $N = 40$. The sampled trajectory still leads to the collision. **(c)** presents the sampled trajectory after replanning with $N = 80$. The trajectory becomes feasible and avoids collision. **(d)** shows the sampled trajectory after replanning with $N = 256$. The trajectory improves quality but needs much more time.

In the supplementary, we first discuss the experimental details and hyperparameters in Section A. Next, we analyze the impact of different numbers of diffusion steps $N$ on the replanning process in Section B, and further present the visualization in RLBench in Section C. Finally, we discuss how to compute the likelihood in Section D.

## A    Experimental Details

1. Our architecture is built based on Diffuser [17] and Decision Diffuser [1]. In detail, our architecture comprises a temporal U-Net structure with six repeated residual networks. Each network consists of two temporal convolutions followed by GroupNorm [39], and a final Mish nonlinearity [26]. Additionally, We incorporate timestep and conditions embeddings, which are both 128-dimensional vectors produced by MLP, within each block.

2. In RLBench, we exploit the image encoder from the pre-trained CLIP [31], followed by a 2-layered MLP with 512 hidden units and Mish activations.

3. The model is trained using Adam optimizer [20] with a learning rate of $2\mathrm{e}{-04}$ and a batch size of 64 for $1\mathrm{e}6$ training steps.

4. The planning horizon is set to 128 in Maze2D//Multi2D U-Maze, 256 in Maze2D//Multi2D Medium, 256 in Maze2D//Multi2D Large, 64 in Stochastic Environments, and 64 in RLBench.

5. We use a threshold of 0.7 for **Replan from scratch** and a threshold of 0.5 for **Replan with future**.

6. The probability $\epsilon$ of random actions is set to 0.03 in Stochastic Environments.

7. The diffusion steps $i$ for computing likelihood is set to $\{5, 10, 15\}$ in Maze2D and Stochastic Environments and $\{10, 20, 30\}$ in RLBench.

8. The total number of diffusion steps, corresponding to the number of diffusion steps for **Replan from scratch** is set to 256 in Maze2D, 200 in Stochastic Environments, and 400 in RLBench. And the number of diffusion steps for **Replan with future** is set to 80 in Maze2D, 40 in Stochastic Environments, and 100 in RLBench.

9. We perform the whole experiment with a total of three Tesla V100 GPUs.

| Environment | Diffuser | RDM ($N = 40$) | RDM (Ours) | RDM ($N = 256$) |
|---|---|---|---|---|
| Multi2D  Large | 129.4 | 160.9 | 195.4 | 197.7 |

Table 4: **Replanning with different diffusion steps.** RDM with $N = 40$ performs better than Diffuser. Our method RDM with $N = 80$ further improves performance and is much more efficient than RDM with $N = 256$.

## B    Different Diffusion Steps for Replanning

In this section, we visualize the impact of using different numbers of diffusion steps $N$ to replan an existing trajectory. Figure 8 illustrates a problematic sampled trajectory after execution. When

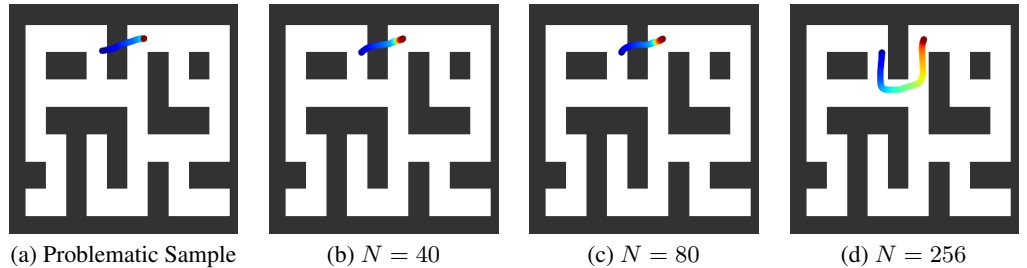

| (a) Problematic Sample | (b) $N = 40$ | (c) $N = 80$ | (d) $N = 256$ |

Figure 9: **Samples of replanning with different diffusion steps. (a)** demonstrates the problematic trajectory. **(b)(c)** presents the sampled trajectory after replanning with $N = 40$ and 80. The previously sampled trajectory is so troublesome that it can't recover from fast replanning. So the sampled trajectories still lead to the collision. **(d)** shows the sampled trajectory after replanning with $N = 256$. After **Replan from scratch**, the resampled trajectory becomes feasible.

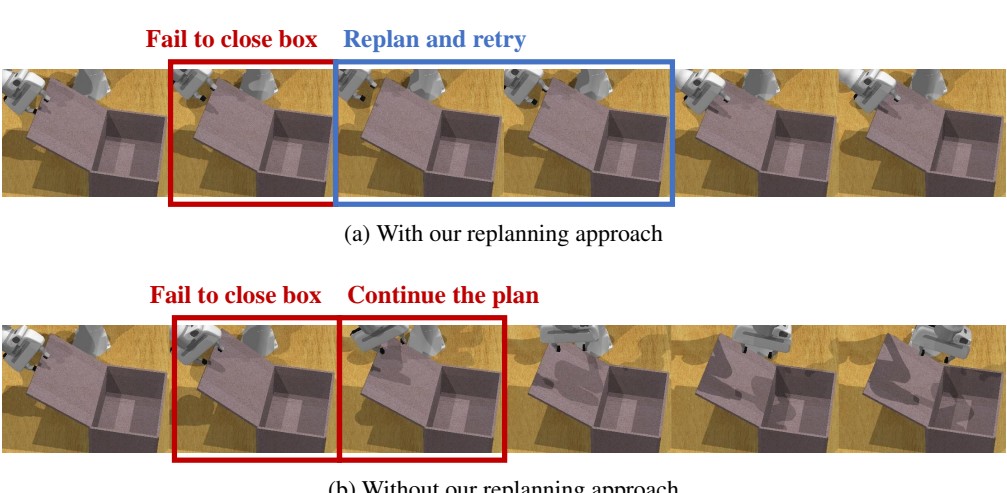

(a) With our replanning approach

(b) Without our replanning approach

Figure 10: **Comparison between trajectories with and without replanning. (a)** Our replanning approach allows the agent to detect when the robotic arm fails to close the box and instructs it to retry. **(b)** Without our replanning approach, the failure goes unnoticed, leading the arm to persist in following the previous, unsuccessful plan.

$N = 40$ of replanning, the resampled trajectory still leads to the collision with the wall. However, with $N = 80$, which aligns with the number of diffusion steps used in **Replan with future**, we observe that the sampled trajectory becomes feasible. For $N = 256$, corresponding to the number of diffusion steps used in **Replan from scratch**, the resampled trajectory shows additional improvement, but such a procedure is substantially more expensive than our replanning procedure.

We illustrate another example plan which based off likelihood would need to be planned from scratch in Figure 9. After replanning with $N = 40$ and $N = 80$, the trajectories still encounter issues with colliding with the wall. However, With $N = 256$, RDM successfully regenerates a completely different and feasible trajectory through replanning from scratch.

We further evaluate the performance with different replanning steps in Table 4. The results clearly demonstrate that that RDM consistently outperforms Diffuser in all cases. Remarkably, RDM with $N = 80$, corresponding to **Replan with future**, achieves performance close to RDM with $N = 256$, which corresponds to **Replan from scratch** which is substantially more expensive.

## C Comparison between trajectories with and without replanning

In this section, we visualize the execution in the task `close box` in Figure 10. In the second frame of Figure 10, we observe that the robotic arm fails to close the box due to insufficient contact between the gripper and the lid. In Figure 10a, RDM successfully detects the failures and retries to establish

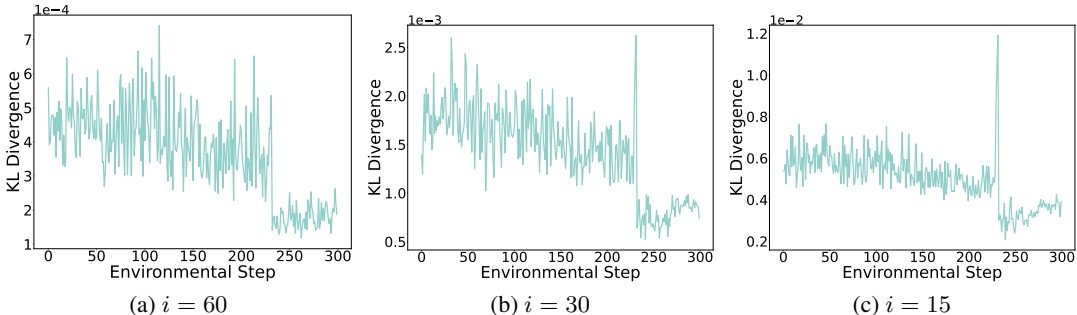

Figure 11: **Different diffusion steps** $i$ **for computing KL divergence.** We estimate the likelihood of trajectory by computing KL divergence between $q(\tau^{i-1} \mid \tau^i, \tau_{\rightarrow k})$ and $p_\theta(\tau^{i-1} \mid \tau^i)$. Here, we illustrate the curve of KL divergence at different environmental steps in Maze2D where the number of total diffusion steps is 256. **(a)** When large amounts of noise are added at $i = 60$, the agent fails to detect any infeasibility as the KL divergence remains low and consistent. **(b)** When intermediate amounts of noise are added at $i = 30$, the agent has the ability to notice infeasibility but the value of KL divergence is similar to that of feasible trajectories. **(c)** When small amounts of noise are added at $i = 15$, the agent can effectively discern infeasibilities, as the KL divergence exhibits a clear boundary between feasible and infeasible trajectories.

proper contact with the lid. As a result, the gripper successfully closes the box this time. In contrast, Figure 10b demonstrates the scenario where the arm fails to notice the failures, making the arm persist in the unsuccessful plan. These results demonstrate the capability of RDM to detect failures and replan a successful trajectory in challenging robotic control tasks.

## D   How to Compute Likelihood

In this section, we analyze the number of noising steps we should use to estimate the likelihood. The likelihood estimation is performed by computing the KL divergence between $q(\tau^{i-1} \mid \tau^i, \tau_{\rightarrow k})$ and $p_\theta(\tau^{i-1} \mid \tau^i)$ with different magnitudes of noise added at diffusion timesteps $i$. We analyze the impact of different diffusion steps $i$ and the corresponding magnitude of noise added for computing KL divergence in Figure 11. We observe that (1) When large amounts of noise are added at $i = 60$ the agent fails to detect any infeasibility, as the KL divergence remains low and consistent. (2) When intermediate amounts of noise are added at $i = 30$, the agent has the ability to notice infeasibilities, but the KL divergence value is similar to that of a feasible trajectory. (3) When small amounts of noise are added at $i = 15$, the agent can effectively discern infeasibilities, as the KL divergence exhibits a clear boundary between feasible and infeasible trajectories. Based on these findings, we choose smaller values of $i$ for computing the likelihood of trajectories and estimating when to replan.

## E   Additional Experimental Results

We show the additional experimental results in the following.

1. We show the results of the comparison between our RDM algorithm and the replanning-based baseline algorithms (SDM and PRDM) across more tasks. We run experiments using different intervals or thresholds and measure their computational costs by the total number of diffusion steps. Notably, RDM consistently outperforms all the baseline algorithms under the same total diffusion steps (that is, with the same computational budget).

2. We investigate different intervals $I$ for replanning for Diffuser and Decision-Diffuser. We observe that as the interval decreases (that is replanning is done more frequently), the performance improves as we expect. However, when the interval is smaller than a certain value, the performance decreases significantly. For example, in the Maze2D Large domain shown in Figure VI, the return increases when the interval decreases from 250 to 100, while it drops when the interval decreases from 100 to 1. The ablation results confirm our statement that replanning with an interval that is too small (for example, replanning at every time step) may prevent successful task execution.

3. We also analyze the impact of different thresholds $l$ for the baseline algorithms, SDM and RDM. As we expect, when $l$ decreases (that is, when replanning is done more frequently), the performance of all the methods has improved. Despite the performance differences of the baseline algorithms, we want to emphasize again that under the same computational budget, our RDM algorithm outperforms all the baseline algorithms.

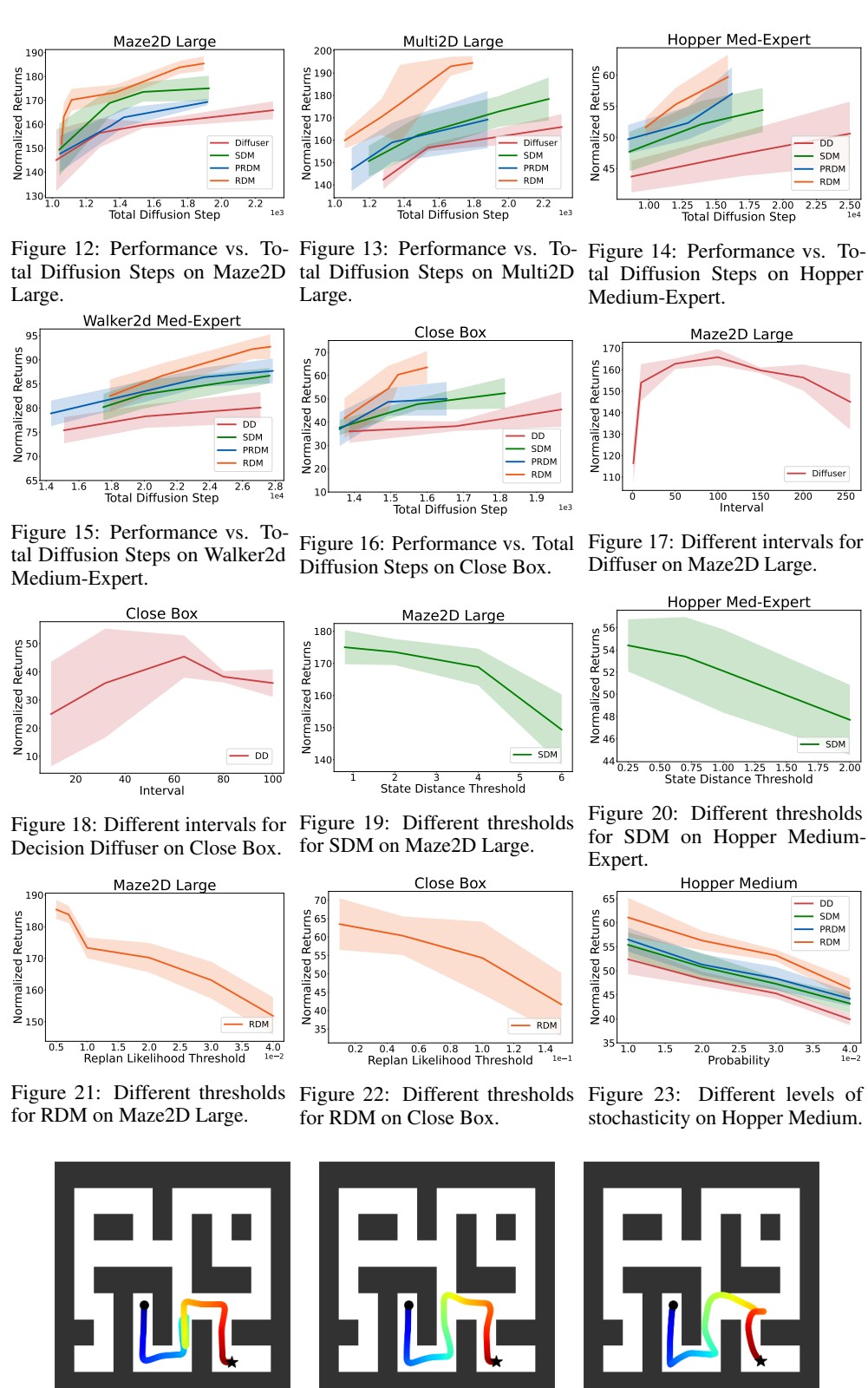

Figure 12: Performance vs. Total Diffusion Steps on Maze2D Large.

Figure 13: Performance vs. Total Diffusion Steps on Multi2D Large.

Figure 14: Performance vs. Total Diffusion Steps on Hopper Medium-Expert.

Figure 15: Performance vs. Total Diffusion Steps on Walker2d Medium-Expert.

Figure 16: Performance vs. Total Diffusion Steps on Close Box.

Figure 17: Different intervals for Diffuser on Maze2D Large.

Figure 18: Different intervals for Decision Diffuser on Close Box.

Figure 19: Different thresholds for SDM on Maze2D Large.

Figure 20: Different thresholds for SDM on Hopper Medium-Expert.

Figure 21: Different thresholds for RDM on Maze2D Large.

Figure 22: Different thresholds for RDM on Close Box.

Figure 23: Different levels of stochasticity on Hopper Medium.

(a) interval=255

(b) interval=50

(c) interval=1

Figure 24: Visualization for Diffuser with different intervals. Replanning too frequently will cause some sub-optimal plans.

