# OpenReview forum: "Adaptive Online Replanning with Diffusion Models"
_NeurIPS.cc/2023/Conference — NeurIPS 2023 poster_

### Official Review · Reviewer_XopQ · 2023-06-20

**Soundness:** 3 good
**Presentation:** 3 good
**Contribution:** 2 fair
**Rating:** 6
**Confidence:** 4

**Summary:**

The authors address the online replanning problem within diffusion-based models by introducing their method, RDM, which uses the likelihood of trajectories to decides when and how to replan.

**Strengths:**

- The topic of when and how to replan is an important one, particularly within the diffusion model community where it has not, to the best of my knowledge, been addressed before.
- The methodology section appears to be sound, and the idea of choosing to replan and what type of replanning should happen based on the trajectory likelihoods is compelling.
- The paper is clear, well-written and easy to follow.

**Weaknesses:**

- The experimental results are lacking in terms of comparison with other simple baselines that are able to replan. While the authors partially address this in the ablation study of Section 4.4, it is unclear why these are also not used in the main experiments. Given the aim of the paper is to provide a good replanning strategy, I do not believe this should be an ablation only, but rather that most results should include “no replanning baselines” (i.e., the ones already provided), and “replanning baselines” as proposed in the ablation. This would allow a reader to take conclusions regarding the actual improvement from performing no replanning to a basic strategy to the proposed method.
- The absence of replanning in the baselines could justify the significant drop of performance in the results in Section 4.1, 4.2 and 4.3. A comparison with replanning baselines (both fully and, for example, with a randomly selected number of steps) strengthens the case that the “when” and “how” to replan are key to the success of the results.
- The paper is also lacking an ablation with a variable number of replanning steps. In Section 4.4 the authors compare DD which replans “using two fixed intervals”, but how does the performance change for a higher replanning frequency?

**Minor comments**:
- Typo in abstract, line 1, should be “have risen as a promising”

**Questions:**

- How does this method compare to a fixed, fully replanning strategy at different intervals in the main scenarios compared in the experiments?
- How are the thresholds in Algorithm 1 tuned, and how does the method perform for different $l_1$/$l_2$ values?

**Limitations:**

The authors address some limitations of their method in Section 6 of the paper, and there are no concerns regarding potential negative societal impact for this work.

---

> ### Author Rebuttal · Authors · 2023-08-10
>
> Dear Reviewer,
>
> We thank the reviewer for the reviews and insightful suggestions.
>
> **Q1. Comparison.**
> > The experimental results are lacking in terms of comparison with other simple baselines that are able to replan.
>
> We show the results of the comparison between our RDM algorithm and the replanning-based baseline algorithms (SDM and PRDM) across more tasks. We run experiments using different intervals or thresholds and measure their computational costs by the total number of diffusion steps. In the table below, we choose one task from each of the three domains and report the total diffusion steps and the performance, and show more evaluation results in Figure I-V in the rebuttal PDF. Notably, RDM consistently outperforms all the baseline algorithms under the same total diffusion steps.
>
> Table A. The total diffusion steps and performance.
>
> |Environment | Method   | Total Diffusion Steps | Normalized Returns |
> | ----------------------------------------- | -------- | --------------------- | ------------------ |
> | Maze2D Large                              | Diffuser | 2304.0                | 165.9 (±3.8)        |
> |                                           | SDM      | 1925.1                | 175.0 (±5.3)        |
> |                                           | PRDM     | 1916.4                | 169.3 (±1.4)        |
> |                                           | RDM      | 1894.38               | 185.4 (±3.0)        |
> | Hopper Medium-Expert                      | DD       | 17100                 | 47.4 (±3.6)         |
> |                                           | SDM      | 18500                 | 54.4 (±3.5)         |
> |                                           | PRDM     | 16200                 | 57 (±4.3)           |
> |                                           | RDM      | 15900                 | 59.7 (±3.8)         |
> | Close Box                                 | DD       | 1968                  | 46.0 (±7.5)         |
> |                                           | SDM      | 1813                  | 52.4 (±6.6)         |
> |                                           | PRDM     | 1653                  | 50 (±7.2)           |
> |                                           | RDM      | 1600                  | 63.5 (±7.0)         |
>
>
> **Q2. Replan interval.**
> > How does the performance change for a higher replanning frequency?
>
> We investigate different intervals $I$ for replanning for Diffuser and Decision-Diffuser. The results are shown in Figure VI and VII. We observe that as the interval decreases (that is replanning is done more frequently), the performance improves as we expect. However, when the interval is smaller than a certain value, the performance decreases significantly. For example, in the Maze2D Large domain shown in Figure VI, the return increases when the interval decreases from $250$ to $100$, while it drops when the interval decreases from $100$ to $1$. The ablation results confirm our statement that replanning with an interval that is too small (for example, replanning at every time step) may prevent successful task execution.
>
> **Q3. Replan threshold.**
> > How are the thresholds in Algorithm 1 tuned, and how does the method perform for different $l_1$/$l_2$ values?
>
> We analyze the impact of different thresholds $l_s$ for RDM. The results are shown in Figure X-XI in the rebuttal PDF. As we expect, when $l$ decreases (that is when replanning is done more frequently), the performance has improved. We choose the best result for different $l_s/l_f$ under the same total diffusion steps.
>
> **Q4. Minor comments.**
> > Typo in abstract, line 1, should be “have risen as a promising”.
>
> Thanks for pointing this out. We will address them in the revision.

---

> > ### Comment · Reviewer_XopQ · 2023-08-14
> > **Response to Rebuttal**
> >
> > Thank you for the detailed rebuttal.
> >
> > **On the comparison**: I thank the authors for the extra results, as they contextualize the performance improvements of RDM. I believe include this in the paper will make the case for the method stronger.
> >
> > **On the replan interval**: the failing to plan with a high replanning frequency is an interesting observation. These results show that a simple fixed time replanning strategy significantly underperforms RDM, which again makes the case for the method stronger.
> >
> > **On the replan threshold for RDM and SDM**: these results are expected, but also showcase the interesting monotonic behavior of the returns based on the threshold. The differences between these returns and the ones obtained by simply reducing the replanning interval also showcases the non-trivial nature of the method.
> >
> > The authors have addressed all the concerns I had regarding the paper, so I am updating the score accordingly.

---

> > > ### Author Response · Authors · 2023-08-16
> > >
> > > Dear reviewer XopQ,
> > >
> > > Thank you for your comments on our response. We are happy that your concerns have been addressed. We still have a few days left in the discussion period. If you have any further questions or if there is anything else we can provide to show the merits of our work, please don't hesitate to let us know. Thank you!
> > >
> > > Best,
> > >
> > > Authors

---

### Official Review · Reviewer_bzBf · 2023-07-05

**Soundness:** 2 fair
**Presentation:** 1 poor
**Contribution:** 2 fair
**Rating:** 5
**Confidence:** 4

**Summary:**

The submission describes a method for deciding when and how to "replan" when using diffusion models for inferring plans, as in Decision Diffuser [1]. The task is essentially imitation learning (IL): given a training dataset of plans and features, the task is to predict a new plan given novel features. As in [1], the IL method consists of training a diffusion model (DM) to predict state sequences given contexts, and the features happen to include the reward of the plan. A high-reward plan can therefore be inferred by conditioning the model on a high reward value.

The replanning aspect of this work is useful because running the diffusion model may be expensive, making it difficult to use diffusion-model-based planning to do receding-horizon control. Replanning only when necessary saves computation compared to planning every cycle.

**Strengths:**

Originality:

The ideas of trying to infer when and how to replan when planning with diffusion models seem novel. I'm not aware of other work on inferring when to replan in general, so that general idea may be novel as well. A nice feature of applying probabilistic methods to planning is that it becomes quite natural to ask such questions.

Quality:

There are some promising aspects to the experiments. I appreciated that the experiments were targeted towards answering specific questions that were clearly articulated in lines 182-185. The robotic control (RLBench) results were also promising in that the success rate was significantly higher than that of a baseline without replanning.

Significance:

The results show that simply replanning based on some simple heuristics is a viable technique for boosting the performance of diffusion-based planning methods. This is a potentially useful and interesting observation for anyone interested in applying diffusion models to solving planning problems in practice.

**Weaknesses:**

Quality:

I believe the weakest part of the submission is that the experiments are focused mainly on "apples-to-oranges" comparisons of RDM to model-free RL and decision diffusion models without replanning. None of the baselines have the ability to respond to new information at test time, as far as I can tell. In the case of the model-free RL algorithms, this is a bit unclear—theoretically, the learned policies could condition on "real-time" information, but it's unclear whether this is the case. Still, even if the RL policies do condition on novel information, comparing a policy learned offline to a planner that can be evaluated online, seems a bit unfair.

It would make more sense to me if the experiments were focused more on comparisons between RDM and other replanning methods. Since the goal of intelligent replanning is to reduce computation with respect to replanning every cycle, I think it would also be fair to evaluate the net effect of different replanning strategies on the tradeoff between solution time and solution quality. For example, if we replan at fixed intervals, but evaluate that for a range of intervals, what trade-off do we see between solution time and quality compared to RDM? If we replan every cycle, is the solution quality much higher than RDM?

Although the ablation study in section 4.4 does analyze the effects of different replanning strategies, this analysis feels unsatisfying for a few reasons. First, it is unclear how the threshold / interval for replanning were chosen for DD and SDM, which simply replan based on time or state distance thresholds. Ideally, these thresholds would be chosen to maximize some metric evaluated on a validation set, but I could find no details about this.

I also note that the baseline for figure 7 is not zero, which gives a very misleading impression about the relative performance gap between (e.g.) SDM (replan triggered by state distance) and RDM, which is actually pretty small. There are also no error bars on this plot, so it is hard to tell whether this result is significant. This experiment was also only run on a single task (hopper-medium-expert)—the results would be much more interesting if these simple baselines (DD, SDM) were run on the entire suite of tasks. If it turns out that SDM performs almost as well as RDM on most tasks, then that would make RDM significantly less appealing.

Clarity:

The problem statement is  unclear. On my initial read-through of the paper, I was under the impression that RDM (the proposed method) was an imitation learning (IL) method—until I reached the experiments, which compared RDM to RL methods. I then backtracked to Section 2.1 (problem setting), which states:

"We consider a trajectory optimization problem similar to Diffuser[17]... the objective of the trajectory optimization problem is to find a sequence of actions … that maximizes J… which is the expected sum of rewards over all timesteps."

After reading the reference for Diffuser[17], this paragraph convinced me that my original belief was wrong, and that RDM is in fact an (online) RL method that uses a diffusion model as a planner—just like Diffuser[17]. I then at some point encountered a Decision Diffuser[1] reference in the paper and decided to revisit that reference to refresh my memory. I was very confused because the paper was not what I remembered it being, and in fact Decision Diffuser[1] is an IL method.

I eventually realized that Diffuser[17] and Decision Diffuser[1] are different methods: the first is an RL method, and the second is an IL method. However, they both share common authors, use similar methods, and have similar names. I then re-read lines 89-91, ("we follow the formulation of planning in Decision Diffuser[1]…") which convinced me that my original impression was correct: RDM is an IL method, not an RL method.

Despite the unfortunately similar names, I believe the real root of my confusion is that the paper lacks a crisp problem statement—the reader should not have to guess as to whether the problem addressed is IL, or offline/online off-policy/on-policy RL. It's unfortunate that Section 2.1 (Problem Setting) introduces the method in RL terms and explicitly states that the problem is similar to that of Diffuser[17], which is an RL method. Instead, that paragraph should probably state something like "Our work addresses a problem similar to that described in Decision Diffuser[1]: given a dataset of training state trajectories and rewards, our task is to predict a state trajectory similar to what was observed in the training dataset, while conditioning on a query reward value. This is essentially an Imitation Learning problem where we observe and condition on rewards, but it is also similar to RL, in the sense that the goal is to produce a trajectory with high reward."

The comparison to RL methods also confuses things—the submission should be clearer about why RDM is compared to RL methods. It would also be beneficial to call Diffuser by a different name (e.g., Planning Diffuser) to disambiguate it from Decision Diffuser.

The exposition of the method could also be significantly improved. The explanation of when to replan in section 3.1 is a bit too verbose and doesn't convey the basic idea well, which is simply to evaluate the likelihood of a plan where states/actions at previous timesteps are replaced with those observed during plan execution. A simple figure would be very helpful here.

Significance:

One factor that may limit the significance of this work is that the topic is relatively niche—it attempts to solve a particular problem (performance of replanning) with applying a particular planning method (Decision Diffuser), which itself is still relatively immature. Solving niche problems is ok, but the potential for significant impact would be greater if there were  some interesting take-away for a more general audience.

**Questions:**

How were the intervals / state distance thresholds set for the DD and SDM baselines in figure 7?

Have you tried evaluating the trade-off between solution quality and planning time for different replanning strategies (e.g., for different possible threshold values in DD and SDM)?

**Limitations:**

The paper adequately addresses the limitations of this work.

---

> ### Author Rebuttal · Authors · 2023-08-10
>
> Dear Reviewer,
>
> We thank the reviewer for the detailed comments and questions.
>
> **Q1. Comparison.**
> > It would make more sense to me if the experiments were focused more on comparisons between RDM and other replanning methods.
> > What trade-off do we see between solution time and quality compared to RDM?
>
> We show the results of the comparison between our RDM algorithm and the replanning-based baseline algorithms across more tasks. We run experiments using different intervals or thresholds and measure their computational time by the total number of diffusion steps. In the table below, we report the total diffusion steps and the performance and show more evaluation results in Figure I-V in the rebuttal PDF. Notably, RDM consistently outperforms all the baseline algorithms under the same total diffusion steps.
>
> Table A. The total diffusion steps and performance.
>
> |Environment|Method|Total Diffusion Steps|Normalized Returns|
> |-----|-----|-----|-----|
> |Maze2D Large|Diffuser|2304.0|165.9 (±3.8)|
> | |SDM|1925.1|175.0 (±5.3)|
> | |PRDM|1916.4|169.3 (±1.4)|
> | |RDM|1894.38|185.4 (±3.0)|
> |Hopper Medium-Expert|DD|17100|47.4 (±3.6)|
> | |SDM|18500|54.4 (±3.5)|
> | |PRDM|16200|57 (±4.3)|
> | |RDM|15900|59.7 (±3.8)|
> |Close Box|DD|1968|46.0 (±7.5)|
> | |SDM|1813|52.4 (±6.6)|
> | |PRDM|1653|50 (±7.2)|
> | |RDM|1600|63.5 (±7.0)|
>
> **Q2. Different replanning strategies.**
> > How the threshold/interval for replanning were chosen for DD and SDM.
> > If we replan every cycle, is the solution quality much higher than RDM?
>
> We investigate different intervals $I$ for replanning for Diffuser and Decision-Diffuser. The results are shown in Figure VI and VII. We observe that as the interval decreases (that is replanning is done more frequently), the performance improves as we expect. However, when the interval is smaller than a certain value, the performance decreases significantly. For example, in the Maze2D Large domain shown in Figure VI, the return drops when the interval decreases from $100$ to $1$. The ablation results confirm our statement that replanning with an interval that is too small (for example, replanning at every time step) may prevent successful task execution.
>
> We also analyze the impact of different thresholds $l$ for the baseline algorithms, SDM and RDM. The results are shown in Figure VIII-XI. As we expect, when $l$ decreases (that is, when replanning is done more frequently), the performance has improved. Despite the performance differences of the baseline algorithms, we want to emphasize again that as shown in Figure I-V, under the same computational budget, our RDM algorithm outperforms all the baseline algorithms.
>
> **Q3. Clarity**
> > The real root of my confusion is that the paper lacks a crisp problem statement.
> > The comparison to RL methods also confuses things.
>
> We apologize that our problem statement might be confusing, we would like to clarify that Diffuser [1], Decision-Diffuser [2], and our work can be seen as following the same setting, which is offline reinforcement learning (offline RL) and generate plans conditioned on maximizing reward (where for Long-Horizon and Robotic Control tasks we use the reward function that corresponds to either reaching the conditioned goal or solving the robotics task respectively). To solve this offline reinforcement learning task, Diffuser, Decision-Diffuser and our work sample a trajectory of states (a plan) that maximizes the conditioned reward, where Diffuser samples from the composition of a diffusion model and value function, while Decision Diffuser and our work sampling from conditional diffusion model conditioned on the desired reward function. This plan is then transformed into a policy.
>
> Since the setting we consider can be considered an offline RL setting, our other baselines are then naturally offline RL algorithms. It's important to note that Decision-Diffuser is not a behavioral cloning method, as in the Mujoco settings, it does not fit a model to all trajectories in a dataset, but rather learns a reward-conditioned trajectory model (similar to Decision Transformer) in order to construct trajectories that maximize reward in an environment.
>
> Our approach towards replanning can be applied to any Diffusion model that synthesizes a trajectory of actions to optimize a reward function and can be applied to either Diffuser or Decision-Diffuser.
>
> We will revise our problem statement in our revision to make this clearer, please let us know if you have any additional questions.
>
> [1] M. Janner, Y. Du, J. B. Tenenbaum, and S. Levine. Planning with diffusion for flexible behavior synthesis. arXiv preprint arXiv:2205.09991, 2022.
>
> [2] A. Ajay, Y. Du, A. Gupta, J. Tenenbaum, T. Jaakkola, and P. Agrawal. Is conditional generative modeling all you need for decision-making? arXiv preprint arXiv:2211.15657, 2022.
>
> **Q4. Significance.**
> > It attempts to solve a particular problem (performance of replanning) with applying a particular planning method (Decision Diffuser)
>
> Thanks for the comment. Our replanning strategy, based off likelihood, can be applied to any diffusion-based planner, which has seen a variety of different applications across different planning settings such as robotics policies [4] and video [5]. Broadly, we believe the idea of likelihood-based replanning can also be applied to many other likelihood-based planning methods (e.g. trajectory transformer[3]).
>
> [3] Michael Janner, Qiyang Li, and Sergey Levine. Reinforcement learning as one big sequence modeling problem. arXiv preprint arXiv:2106.02039, 2021.
>
> [4] Cheng Chi, Siyuan Feng, Yilun Du, Zhenjia Xu, Eric Cousineau, Benjamin Burchfiel, Shuran Song. Diffusion Policy: Visuomotor Policy Learning via Action Diffusion. arXiv preprint arXiv:2303.04137, 2023.
>
> [5] Yilun Du, Mengjiao Yang, Bo Dai, Hanjun Dai, Ofir Nachum, Joshua B. Tenenbaum, Dale Schuurmans, Pieter Abbeel. Learning Universal Policies through Text-Conditioned Video Generation. arXiv preprint arXiv:2302.00111, 2023.

---

> > ### Author Response · Authors · 2023-08-16
> >
> > Dear reviewer bzBf,
> >
> > Thank you again for your comments and suggestions on our paper. We hope that our responses and new results have addressed your questions and concerns. We still have a few days left in the discussion period. If you have any further questions, please don't hesitate to let us know and we'll be happy to address them. Thank you!
> >
> > Best,
> >
> > Authors

---

> > > ### Comment · Reviewer_bzBf · 2023-08-17
> > > **Rebuttal response**
> > >
> > > Thank you for adding the results comparing the total diffusion steps and task performance across different methods. This does alleviate a significant concern of mine.

---

> > > > ### Author Response · Authors · 2023-08-17
> > > >
> > > > Dear reviewer bzBf,
> > > >
> > > > Thank you for reviewing our new results and we are glad that they have resolved your concerns. We would appreciate it if you could increase your score! Please don’t hesitate to let us know if there is anything else we can provide.
> > > >
> > > > Best,
> > > >
> > > > Authors

---

> > > > > ### Author Response · Authors · 2023-08-19
> > > > > **Follow-up on score updates!**
> > > > >
> > > > > Dear reviewer bzBf,
> > > > >
> > > > > As the rebuttal period is ending soon, we wonder if our response answers your questions and addresses your concerns. If yes, would you kindly consider raising the score? Thanks again for your very constructive and insightful feedback!
> > > > >
> > > > >
> > > > > Best,
> > > > >
> > > > > Authors

---

> > > > ### Comment · Area_Chair_FGjD · 2023-08-19
> > > >
> > > > Hi,
> > > >
> > > > Thank you for your help reviewing this paper and for participating in the discussion with the authors.
> > > >
> > > > It seems that the authors' response resolves resolves at least some of the issues that you raised in your initial review. Please be sure to update your review and rating accordingly.
> > > >
> > > > Best,\
> > > > AC

---

> > > > > ### Author Response · Authors · 2023-08-21
> > > > >
> > > > > Dear reviewer bzBf,
> > > > >
> > > > > As the rebuttal period is ending today, we wonder if our response answers your questions and addresses your concerns. If yes, would you kindly consider raising the score? Thanks again for your very constructive and insightful feedback!
> > > > >
> > > > > Best,
> > > > >
> > > > > Authors

---

### Official Review · Reviewer_rt81 · 2023-07-05

**Soundness:** 3 good
**Presentation:** 3 good
**Contribution:** 2 fair
**Rating:** 5
**Confidence:** 4

**Summary:**

The manuscript introduces a technique for enhance the motion planner rooted in diffusion models, encompassing decisions concerning the timing of replanning and plan trajectories upon the existing path. The strategy for timing replanning utilizes the inherent estimated likelihood of the trajectory in diffusion models as a criterion for replanning. In terms of forming new trajectories, the authors either completely reconstruct the plan or modify it based on future contexts determined by specific guidelines. The concept is clear-cut, and the outcomes appear to be quite promising.

**Strengths:**

Replanning is important for robots to execute trajectories robustly, deciding when to replan and replanning algorithm are the core challenges of replanning problem. Tackling these problems can help a lot with diffusion based robot motion planning methods. Moreover, the overall writing quality is good, and the core idea is presented well.

**Weaknesses:**

It's not surprise that the performance gets better by adding replanning strategy into diffusion model. For long horizon planning problem in Section 4.1, the authors compared performance of RDM and baselines, however, none of the baselines contains replanning strategy, making the effectiveness of replanning timing decision method proposed by the author hard to evaluate. This also happens to Robotic Control tasks in Section 4.3.

As for stochastic environment in Section 4.2, the author demonstrates that RDM outperforms baseline models in environment with stochastic transition models. How different levels of stochasticity will affect the performance is later proved in Section 4.4. However, only one environment is tested, and the performance of RDM and other baselines are relatively close, which is inadequate to evaluate the effectiveness of replanning strategy in RDM.

In Section 4.4, the author compares the performance for models using different replanning strategy, and comparison of different fixed interval/different threshold of state distance deviation is missing, which makes the performance of DD/SDM not convincing enough.

There are also some points which are not demonstrated very well:
- The detailed implementation of how the replanned trajectory is generated is missing.
- The theoretical proof of the effectiveness of likelihood function for partially-executed trajectory is missing, and the threshold of likelihood to choose how to replan is selected empirically which may significantly impact the performance.

**Questions:**

- For long horizon planning tasks and robotic control tasks, is using likelihood better than using state distance deviation as replanning criteria?
- Comparing baselines with different replanning timing decision methods and RDM, how much efficiency improvement will RDM achieve?
- For stochastic environment, the advantage of methods implementing replanning strategy comparing to non-replan methods should be increased when stochasticity arises, but the performance gets closer shown on Figure 6. Why?

**Limitations:**

The author addressed the limitations well.

---

> ### Author Rebuttal · Authors · 2023-08-10
>
> Dear Reviewer,
>
> We thank the reviewer for the detailed reviews and insightful suggestions.
>
> **Q1. Replanning baselines.**
> > For long horizon planning tasks and robotic control tasks, is using likelihood better than using state distance deviation as replanning criteria?
> >
> > Comparing baselines with different replanning timing decision methods and RDM, how much efficiency improvement will RDM achieve?
>
> We show the results of the comparison between our RDM algorithm and the replanning-based baseline algorithms (SDM and PRDM) across more tasks. We run experiments using different intervals or thresholds and measure their computational costs by the total number of diffusion steps. In the table below, we choose one task from each of the three domains and report the total diffusion steps and the performance, and show more evaluation results in Figure I-V in the rebuttal PDF. Notably, RDM consistently outperforms all the baseline algorithms under the same total diffusion steps.
>
> Table A. The total diffusion steps and performance.
> |Environment | Method   | Total Diffusion Steps | Normalized Returns |
> | ----------------------------------------- | -------- | --------------------- | ------------------ |
> | Maze2D Large                              | Diffuser | 2304.0    | 165.9 (±3.8)        |
> |        | SDM      | 1925.1        | 175.0 (±5.3)        |
> |         | PRDM     | 1916.4                | 169.3 (±1.4)        |
> |              | RDM      | 1894.38               | 185.4 (±3.0)        |
> | Hopper Medium-Expert                      | DD       | 17100        | 47.4 (±3.6)         |
> |          | SDM      | 18500                 | 54.4 (±3.5)         |
> |             | PRDM     | 16200                 | 57 (±4.3)           |
> |                | RDM      | 15900                 | 59.7 (±3.8)         |
> | Close Box                                 | DD       | 1968                  | 46.0 (±7.5)         |
> |            | SDM      | 1813                  | 52.4 (±6.6)         |
> |                 | PRDM     | 1653                  | 50 (±7.2)           |
> |              | RDM      | 1600                  | 63.5 (±7.0)         |
>
>
> **Q2. Stochasticity levels.**
> > However, only one environment is tested, and the performance of RDM and other baselines are relatively close, which is inadequate to evaluate the effectiveness of replanning strategy in RDM.
>
> We show the comparative results about different levels of stochasticity in Figure XII in our rebuttal PDF. The reason why all planning-based methods do not perform very well is that randomness will sometimes make the agent reach the out-of-distribution states, which leads to the performance drop.
>
> **Q3. Unclear points.**
> > The detailed implementation of how the replanned trajectory is generated is missing.
>
> We have presented how to replan in Section 3.2 and provided the pseudocode in Algorithm 3. It requires a partially executed plan, $\tilde{\tau}$, as input. It adds noises to it by running the forward process of the diffusion model for $N_f$ steps (where $N_f$ is a pre-defined parameter), and then denoise the trajectory by running the denoising step of the diffusion model for $N_f$ steps. We will make the implementation clearer in the revision.
>
> > The theoretical proof of the effectiveness of likelihood function for partially-executed trajectory is missing, and the threshold of likelihood to choose how to replan is selected empirically which may significantly impact the performance.
>
> We investigate the different thresholds in Figure X and XI and empirically show that our model performs better given the same total diffusion steps.
>
> Also note that our paper is focused on *empirically* validating the effectiveness of using the likelihood function to determine when to replan. We do not aim to claim that this method is theoretically-optimal or the return of the replaned trajectories has any theoretical guarantees, but do note that replanning based off likelihood is principled as it occurs precisely when states / plans fall outside the distribution learned by the original trajectory model.
>
> **Q4. Stochastic environment.**
> > For stochastic environment, the advantage of methods implementing replanning strategy comparing to non-replan methods should be increased when stochasticity arises, but the performance gets closer shown on Figure 6. Why?
>
> The reason is that randomness will sometimes cause the agent to reach the out-of-distribution states, which leads to the drop of the performance of all offline RL methods. Our method RDM consistently outperforms other baselines under different levels of stochasticity. However, at high stochasticity levels, all methods will tend to fail.

---

> > ### Author Response · Authors · 2023-08-16
> >
> > Dear reviewer rt81,
> >
> > Thank you again for your comments and suggestions on our paper. We hope that our responses and new results have addressed your questions and concerns. We still have a few days left in the discussion period. If you have any further questions, please don't hesitate to let us know and we'll be happy to address them. Thank you!
> >
> > Best,
> >
> > Authors

---

> > ### Comment · Reviewer_rt81 · 2023-08-19
> > **Rebuttle Response**
> >
> > Thank you for taking the time to provide a thorough explanation. It's evident that the RDM replanning method holds an advantage over more basic replanning techniques, especially in longer sequence tasks. However, its performance in environments with high levels of stochasticity seems to be an area of potential improvement. While optimizing the timing of replanning via likelihood function to enhance the planning success rate is a valuable insight, it doesn't drastically alter my initial impression. Nonetheless, I truly appreciate your efforts in clarifying the methodology.

---

### Official Review · Reviewer_kWoD · 2023-07-10

**Soundness:** 3 good
**Presentation:** 3 good
**Contribution:** 3 good
**Rating:** 5
**Confidence:** 4

**Summary:**

This paper introduces Replanning with Diffusion Models (RDM), which utilizes an internally estimated likelihood of the current plan to determine when to perform replanning. The authors propose various strategies for replanning in different scenarios.

**Strengths:**

1. The introduction of Replanning with Diffusion Models (RDM) and the use of an internally estimated likelihood of the current plan for replanning strategies is novel.
2. The experimental results demonstrate impressive performance in robot control and stochastic environments, surpassing baselines like IQL.

**Weaknesses:**

1. The validation of the proposed method is not fully comprehensive. In RDM, a key aspect is the estimation of likelihood, but the authors did not discuss the accuracy of the estimation or its impact on final performance.
2. The paper contains some language issues and typos that should be carefully reviewed. For example, in line 178, it should refer to Figure 7 instead of Table 7. Additionally, in line 37, "we propose a principled approach to xxx" has a grammatical error.
3. In the robot control experiments using the RLBench domain, comparing IQL or DT as baselines might not be sufficient. It would be beneficial to compare against other agents known for performing well in the RLBench domain.

**Questions:**

1. I am concerned about the computational resource requirements introduced by the Diffusion Models. For achieving similar performance levels, could you provide insights into the training time and resources needed for IQL, DT, and RDM?
2. While I understand the advantages of replanning, which reduce unnecessary exploration, in RL learning, accurate judgments cannot be made without visiting certain states. How can you ensure that the states not visited after replanning are indeed redundant for learning?
3. The direct introduction of noise into the D4RL dataset (Sec. 4.2) seems unreasonable. Why not collect data directly from random environments to validate this point?

---

> ### Author Rebuttal · Authors · 2023-08-10
>
> Dear Reviewer,
>
> We thank the reviewer for the comments and insightful suggestions.
>
> **Q1. Estimation of likelihood.**
> > In RDM, a key aspect is the estimation of likelihood, but the authors did not discuss the accuracy of the estimation or its impact on final performance.
>
> The accuracy of the estimation of likelihood depends on the number of diffusion steps to compute the likelihood. We run 3 diffusion steps in the experiment. In the table below, we compare the performance under different diffusion steps of computing. We find that 3 steps are sufficient to estimate the likelihood accurately. While using more diffusion steps indeed improves the normalized returns, the improvement is marginal.
>
> Table B. Comparison under different diffusion steps of computing likelihood.
> | Different diffusion steps | 1 | 3 (Ours) |     9   | 15 |
> | --------- | --------- | --------- | --------- | --------- |
> | Normalized Returns    | 179.1 (±4.9) | 185.4 (±3.0)  | 187.0 (±4.0) | 187.2 (±2.2) |
>
> **Q2. RLBench baselines.**
> > It would be beneficial to compare against other agents known for performing well in the RLBench domain.
>
> Thanks for the suggestion. Note that our method plans using preliminary actions in RLBench. As we have discussed in Line 233-238 in the paper, most other algorithms evaluated on RLBench use *micro-steps*, which do not plan using preliminary actions and cannot be directly compared with our method. To the best of our knowledge, IQL and DT are state-of-the-art offline RL algorithms that plan using preliminary actions in RLBench. So we use them as baselines in our paper.
>
> **Q3. Training time.**
> > For achieving similar performance levels, could you provide insights into the training time and resources needed for IQL, DT, and RDM?
>
> Thanks for the suggestion. We list the training time in the table below and each model is trained with only one Tesla V100 GPU. It's hard to report the training time with exactly the same performance level. So we instead report the performance under similar training times in the table below. Our model with 4 hours of training time still outperforms IQL and DT.
>
> Table C. Training Time of all models on Close Box.
> | Model           | IQL  | DT    | RDM*       | RDM          |
> |-----------------|------|-------|-----------|--------------|
> | Training Time (hours)   | 3.3 | 4    | 4        | 8           |
> | Performance     | 10.3 | 7.8   | 58.3 (±8.3)| 61.5 (±7.7)   |
>
> \* We use an earlier checkpoint that has a similar computation time to the baseline algorithms.
>
>
> **Q4. Exploration.**
> > How can you ensure that the states not visited after replanning are indeed redundant for learning?
>
> Thanks for the insightful question. We can agree that in general, we can not ensure that the states not visited will be redundant for learning. One way to integrate exploration into our planning procedure is to measure the uncertainty of the state in a similar way to RND [1] and then determine the timing for replanning based on both the likelihood and the uncertainty of the unvisited states. We leave it as future work and will add this to the discussion of the paper.
>
> [1] Yuri Burda, Harrison Edwards, Amos Storkey, and Oleg Klimov. Exploration by random network distillation. arXiv preprint arXiv:1810.12894, 2018.
>
>
> **Q5. Stochastic Environments.**
> > The direct introduction of noise into the D4RL dataset (Sec. 4.2) seems unreasonable. Why not collect data directly from random environments to validate this point?
>
> We believe that in practice, in many environments, there is unexpected noise in control, due to either an inaccurate controller or external environment perturbances and believe that the addition of stochasticity into the D4RL environment serves to represent this. We are happy to run additional evaluations on an environment with random environments in the final version of the paper.
>
>
> **Q6. Language issues and typos.**
> > For example, in line 178, it should refer to Figure 7 instead of Table 7. Additionally, in line 37, "we propose a principled approach to xxx" has a grammatical error.
>
> Thanks for pointing these out. We will address them in the revision.

---

> > ### Author Response · Authors · 2023-08-16
> >
> > Dear reviewer kWoD,
> >
> > Thank you again for your comments and suggestions on our paper. We hope that our responses and new results have addressed your questions and concerns. We still have a few days left in the discussion period. If you have any further questions, please don't hesitate to let us know and we'll be happy to address them. Thank you!
> >
> > Best,
> >
> > Authors

---

### Official Review · Reviewer_G16t · 2023-07-17

**Soundness:** 3 good
**Presentation:** 3 good
**Contribution:** 3 good
**Rating:** 6
**Confidence:** 4

**Summary:**

This paper studies how to effectively replan with diffusion models. The authors propose an adaptive online replanning strategy using diffusion models. This strategy employs the estimated likelihood of a plan's success to determine when replanning is needed, avoiding frequent, computationally expensive replanning. It also ensures new plans align with the original trajectory's goal, leveraging previously generated plans efficiently. This method led to a 38% performance improvement over previous diffusion planning approaches on Maze2D and enabled the handling of stochastic and long-horizon robotic control tasks.

Main Contributions:

1. A method to determine when to replan with diffusion
2. A method to generate new plans while utilizing the existing plan

**Strengths:**

- **Movitation**: Online re-planning is a natural and important complement to many planning methods, thus the motivation of this work is important.

- **Methodology**: This paper proposes two methods: one is used to determine when to replan while the other one is for how to replan. The two methods worked together to conduct effective online replanning with diffusion models. As far as I know, the proposed methods are novel.

- **Performance Improvement**: The proposed method significantly improves the performance of diffusion planners, with a reported 38% gain over past diffusion planning approaches on Maze2D. And the authors also showcased that the proposed method enable the handling of stochastic and long-horizon robotic control tasks.

**Weaknesses:**

- It is still unclear to me why replaning at every time step does not work. In the figure 7, the Dcesion Diffuser replans at fixed intervals and it performs much worse than the proposed method. However, the authors did not mention how large is the interval is. If the replaning frequency is very low than it should apprently works worse than the proposed method. And in the abstract, the author mentioned that "replanning at each timestep may prevent successful task execution, as different generated plans prevent consistent progress to any particular goal". I am not convinced by this argument without any exmaples or further explanations.

- It is also unclear to me why "Replan on State Distance (SDM)" works so badly according to Figure 7. In Section 3.1, the authors mentioned that "However, in many cases, even if the actual state the agent is in matches with the state in the plan, new information about the environment may make the current plan infeasible." This does not make too much sense to me. I appreciate it if the authors could provide further details about this point.

- Computation budget is an important factor to consider when comparing different methods. (If we have an unlimited computation budget, I feel it is a good idea to replan at each timestep.) However, the numbers of replanning used in each baseline are not present in the paper. If the numbers of replanning vary a lot among different methods, then I would not think it is a fair comparison. I wish the authors could provide the numbers of replanning used in each method.

- Although the proposed method aims to avoid frequent, computationally expensive replanning, it's not clear how computationally efficient the new method is, and whether the computational cost of determining when to replan offsets the benefits gained from less frequent replanning.

**Questions:**

- It would be great to provide the number of "replan from scratch" and the number of "replan with future context" used in each experiment, since this would help the audience further understand how the replanning helps the agent.
- Why does "replan with future context" work better than "replan from previous context"? Could you give any intuitive explanations?
- See the weakness section for other questions.

**Limitations:**

- This method only applies to diffusion models as it adds noises at some diffusion steps. It is unclear how to utilize this method in other planning methods.
- See the weakness section for other limitations.

---

> ### Author Rebuttal · Authors · 2023-08-10
>
> Dear Reviewer,
>
> We appreciate the reviewer for the detailed comments and insightful suggestions.
>
> **Q1. About replanning at every time step.**
> > It is still unclear to me why replaning at every time step does not work.
>
> We investigate different intervals $I$ for replanning for Diffuser and Decision-Diffuser. The results are shown in Figure VI and VII. We observe that as the interval decreases (that is replanning is done more frequently), the performance improves as we expect. However, when the interval is smaller than a certain value, the performance decreases significantly. For example, in the Maze2D Large domain shown in Figure VI, the return increases when the interval decreases from $250$ to $100$, while it drops when the interval decreases from $100$ to $1$. The ablation results confirm our statement that replanning with an interval that is too small (for example, replanning at every time step) may prevent successful task execution.
>
>
> **Q2. Performance of SDM.**
> > It is also unclear to me why "Replan on State Distance (SDM)" works so badly according to Figure 7.
>
> Thanks for the question. In our rebuttal PDF, we compare RDM and SDM in most tasks in Figure I-V and observe that RDM performs better than SDM given the same total diffusion steps. The reason might be that the distance is only computed on a state level, not on a trajectory level. For example, in Figure 1 of the paper, the agent follows the initial plan perfectly. However, it finds the first door inaccessible only when it reaches in front of the door. Although the current observed state is close to the planned state, it still needs to replan as the current plan is infeasible.
>
> Another example is shown in Figure 5. The agent tries to open a box, but it fails to open the box although it follows the initial plan perfectly. In this case, the state distance between the current state and the planned state is still small. If we use Replan on State Distance (SDM), the agent would not replan. On the other hand, RDM computes the likelihood based on the current environment observation and finds that the likelihood of the original plan is low (since the box doesn't open). It will replan although the current state appears similar to the planned state.
>
>
> **Q3. Computation budget (the number of replanning).**
> > I wish the authors could provide the numbers of replanning used in each method.
>
> > It would be great to provide the number of "replan from scratch" and the number of "replan with future context" used in each experiment
>
> Thanks for the suggestion. We show the results of the comparison between our RDM algorithm and the replanning-based baseline algorithms (SDM and PRDM) across more tasks. We run experiments using different intervals or thresholds and measure their computational costs by the total number of diffusion steps. In the table below, we choose one task from each of the three domains and report the total diffusion steps and the performance, and show more evaluation results in Figure I-V in the rebuttal PDF. Notably, RDM consistently outperforms all the baseline algorithms under the same total diffusion steps (that is, with the same computational budget).
>
> Table A. The total diffusion steps and performance.
> |Environment | Method   | Total Diffusion Steps | Normalized Returns |
> | ----------------------------------------- | -------- | --------------------- | ------------------ |
> | Maze2D Large             | Diffuser | 2304.0     | 165.9 (±3.8)        |
> |          | SDM      | 1925.1   | 175.0 (±5.3)        |
> |          | PRDM     | 1916.4     | 169.3 (±1.4)        |
> |      | RDM      | 1894.38      | 185.4 (±3.0)        |
> | Hopper Medium-Expert                      | DD       | 17100   | 47.4 (±3.6)         |
> |        | SDM      | 18500    | 54.4 (±3.5)         |
> |     | PRDM     | 16200     | 57 (±4.3)           |
> |      | RDM      | 15900       | 59.7 (±3.8)         |
> | Close Box              | DD       | 1968      | 46.0 (±7.5)         |
> |    | SDM      | 1813    | 52.4 (±6.6)         |
> |      | PRDM     | 1653           | 50 (±7.2)           |
> |     | RDM      | 1600          | 63.5 (±7.0)         |
>
> **Q4. Replan with future context.**
> > Why does "replan with future context" work better than "replan from previous context"? Could you give any intuitive explanations?
>
> Thanks for the question. "Replan from preview context" serves as a baseline method to confirm that, during the execution of a plan, replanning by conditioning on the past states does not help generate better plans. On the other hand, including only the future states starting from the current state to the input to the diffusion model (our design choice in "replan with future context") generates the best plan. The reason might be that past states can be distracting to the diffusion model and do not help generate a better plan.
>
> **Q5. Limitations.**
> > This method only applies to diffusion models as it adds noises at some diffusion steps. It is unclear how to utilize this method in other planning methods.
>
> Our replanning strategy, based off likelihood, can be applied to any diffusion-based planner, which has seen a variety of different applications across different planning settings such as robotics policies [1] and video [2]. Broadly, we believe the idea of likelihood-based replanning can also be applied to many other likelihood-based planning methods (e.g. trajectory transformer[3]).
>
> [1] Cheng Chi, Siyuan Feng, Yilun Du, Zhenjia Xu, Eric Cousineau, Benjamin Burchfiel, Shuran Song. Diffusion Policy: Visuomotor Policy Learning via Action Diffusion. arXiv preprint arXiv:2303.04137, 2023.
>
> [2] Yilun Du, Mengjiao Yang, Bo Dai, Hanjun Dai, Ofir Nachum, Joshua B. Tenenbaum, Dale Schuurmans, Pieter Abbeel. Learning Universal Policies through Text-Conditioned Video Generation. arXiv preprint arXiv:2302.00111, 2023.
>
> [3] Michael Janner, Qiyang Li, and Sergey Levine. Reinforcement learning as one big sequence modeling problem. arXiv preprint arXiv:2106.02039, 2021.

---

> > ### Author Response · Authors · 2023-08-16
> >
> > Dear reviewer G16t,
> >
> > Thank you again for your comments and suggestions on our paper. We hope that our responses and new results have addressed your questions and concerns. We still have a few days left in the discussion period. If you have any further questions, please don't hesitate to let us know and we'll be happy to address them. Thank you!
> >
> > Best,
> >
> > Authors

---

> > ### Comment · Reviewer_G16t · 2023-08-18
> > **Thanks for the rebuttal!**
> >
> > Thank the authors for the rebuttal, and some of my concerns have been addressed. See the follow-up questions below.
> >
> > - **About replanning at every time step:** What happens if the interval is very small? Can you provide an intuitive example?
> > - **Computation cost of determining when to replan:** This is the 4th point I raised in my initial review, I wish the authors could provide some details about it.

---

> > > ### Author Response · Authors · 2023-08-18
> > > **Thanks for your response**
> > >
> > > Dear Reviewer:
> > >
> > > Thank you for the insightful questions.
> > >
> > > **Q1. About replanning at every time step.**
> > >
> > > We visualized the agent's trajectories under different replanning intervals in Figure XIII in the rebuttal PDF. We can see that if the replanning interval is very small (in Figure XIII \(c\), the agent replans at every time step), the agent appears "staggering" in the environment. This is because the plans generated by different environmental steps may be inconsistent. This observation confirms that replanning at every time step would prevent consistent progress towards the goal, which results in a worse performance.
> > >
> > > **Q2. Computation cost of determining when to replan.**
> > >
> > > The computation cost of RDM depends on the number of diffusion steps for planning and replanning. The number of diffusion steps for planning and replanning is about $2 \times 10^4$ while the number for determining when to replan is about $3 \times 10^3$ in stochastic environments. And for long-horizon planning, the numbers are about $2 \times 10^3$ for planning and replanning and $2 \times 10^2$ for determining when to replan.
> > >
> > > Kindly let us know if our responses have addressed your questions and concerns, and if you have further questions. Thank you!
> > >
> > > Best,
> > >
> > > Authors

---

> > > > ### Comment · Reviewer_G16t · 2023-08-18
> > > >
> > > > Thank you for the response. I don't have further questions for now. I have revised my rating accordingly.

---

### Author Rebuttal · Authors · 2023-08-10

We thank all the reviewers for taking the time to review our paper and providing insightful and detailed feedback. We appreciate that the reviewers recognize the following contributions.

* **Significance of our problem**. Replanning is a crucial problem in many planning problems.
> Online re-planning is a natural and important complement to many planning methods. (Reviewer G16t)
>
> Replanning is important for robots to execute trajectories robustly. (Reviewer rt81)
>
> This is a potentially useful and interesting observation. (Reviewer bzBf)
>
> The topic of when and how to replan is an important one. (Reviewer XopQ)

* **Methodology**. Our approach to determining when and how to replan is novel and compelling.
> As far as I know, the proposed methods are novel. (Reviewer G16t)
>
> Replanning strategies is novel. (Reviewer kWoD)
>
> The ideas of trying to infer when and how to replan seem novel. (Reviewer bzBf)
>
> The methodology section appears to be sound, and compelling. (Reviewer XopQ)

* **Performance**. Our approach brings significant improvement.
> The proposed method significantly improves the performance of diffusion planners. (Reviewer G16t)
>
> The experimental results demonstrate impressive performance surpassing baselines like IQL. (Reviewer kWoD)
>
> There are some promising aspects to the experiments. (Reviewer bzBf)

* **Presentation**. Some reviewers acknowledge the clear presentation of the paper.
> The overall writing quality is good, and the core idea is presented well. (Reviewer rt81)
>
> The paper is clear, well-written and easy to follow. (Reviewer XopQ)

We want to emphasize again that the **main contributions** of our work are as follows.
* We propose a principled and novel approach to determine when and how a diffusion model should replan in planning tasks.
* We empirically validate the effectiveness of our algorithm in various domains, use comprehensive ablative studies to justify the design of our algorithm, and demonstrate that our algorithm outperforms other state-of-the-art offline RL algorithms in these domains.

In the rebuttal, we address the reviewers' questions and concerns by providing the following new results in our responses and in the rebuttal PDF as well as reviewer-specific comments in the per reviewer response.

**Comprehensive evaluations of the baseline algorithms on more domains.**

* **Replanning-based Baselines**. We show the results of the comparison between our RDM algorithm and the replanning-based baseline algorithms (SDM and PRDM) across more tasks. We run experiments using different intervals or thresholds and measure their computational costs by the total number of diffusion steps. In the table below, we choose one task from each of the three domains and report the total diffusion steps and the performance, and show more evaluation results in Figure I-V in the rebuttal PDF. Notably, RDM consistently outperforms all the baseline algorithms under the same total diffusion steps (that is, with the same computational budget).

Table A. The total diffusion steps and performance.
|Environment | Method   | Total Diffusion Steps | Normalized Returns |
| ----------------------------------------- | -------- | --------------------- | ------------------ |
| Maze2D Large                              | Diffuser | 2304.0                | 165.9 (±3.8)        |
|                                           | SDM      | 1925.1                | 175.0 (±5.3)        |
|                                           | PRDM     | 1916.4                | 169.3 (±1.4)        |
|                                           | RDM      | 1894.38               | 185.4 (±3.0)        |
| Hopper Medium-Expert                      | DD       | 17100                 | 47.4 (±3.6)         |
|                                           | SDM      | 18500                 | 54.4 (±3.5)         |
|                                           | PRDM     | 16200           | 57 (±4.3)           |
|                                           | RDM      | 15900                 | 59.7 (±3.8)         |
| Close Box                                 | DD       | 1968                  | 46.0 (±7.5)         |
|                                           | SDM      | 1813                  | 52.4 (±6.6)         |
|                                           | PRDM     | 1653                  | 50 (±7.2)           |
|                                           | RDM      | 1600                  | 63.5 (±7.0)         |


**Comprehensive ablative studies for diffusion-based algorithms.**

* **Different Intervals for Replanning**. We investigate different intervals $I$ for replanning for Diffuser and Decision-Diffuser. The results are shown in Figure VI and VII. We observe that as the interval decreases (that is replanning is done more frequently), the performance improves as we expect. However, when the interval is smaller than a certain value, the performance decreases significantly. For example, in the Maze2D Large domain shown in Figure VI, the return increases when the interval decreases from $250$ to $100$, while it drops when the interval decreases from $100$ to $1$. The ablation results confirm our statement that replanning with an interval that is too small (for example, replanning at every time step) may prevent successful task execution.
* **Different Thresholds for Replanning**. We also analyze the impact of different thresholds $l$ for the baseline algorithms, SDM and RDM. The results are shown in Figure VIII-XI. As we expect, when $l$ decreases (that is, when replanning is done more frequently), the performance of all the methods has improved. Despite the performance differences of the baseline algorithms, we want to emphasize again that as shown in Figure I-V, under the same computational budget, our RDM algorithm outperforms all the baseline algorithms.

Please see our detailed responses to all the reviewers below.

---

### Author Response · Authors · 2023-08-12
**Thank you and we are looking forward to your post-rebuttal feedback!**

Dear AC and all reviewers:

Thanks again for all the insightful comments and advice, which helped us improve the paper's quality and clarity.

The discussion phase has been on for several days and we have not heard any post-rebuttal responses yet.

We would love to convince you of the merits of the paper. Please do not hesitate to let us know if there are any additional experiments or clarification that we can offer to make the paper better. We appreciate your comments and advice.

Best,

Author

---

### Author Response · Authors · 2023-08-18
**Looking forward to your post-rebuttal feedback!**

Dear AC:

Thanks again for handling our submission efficiently!

Till now, we are still waiting for the feedback from Reviewer G16t, kWoD and rt81.

We fully understand the reviewers might be too busy to respond timely, but we do hope for more discussions and convince them of the merit of our work!

Could you please help us reach out to the reviewers for feedback?

We appreciate your kind support!

Best,

Authors

---

### Decision · Program_Chairs · 2023-09-21

**Decision:**

Accept (poster)

**Comment:**

The paper considers the problem of replanning in settings where diffusion models are used for planning, a problem that is becoming increasinginly important and yet has received very little attention of-late. The paper proposes a method (RDM) that reasons over when to initiate replanning (based on the diffusion models estimated likelihood of generated plans) and how to perform replanning when the underlying planner is diffusion-based.

The paper was evaluated by five reviewers who largely agree that the problem of online replanning in the case of diffusion model-based planning is particularly important. They also find that the proposed framework, which involves separate methods for determining when and how to replan, is novel and shown to yield significant performance gains. However, the reviewers raised several questions/concerns regarding the computational cost of the method and deficiencies in the evaluation, including the need to compare to more apples-to-apples baselines. The authors made a concerted effort to resolve the reviewers' questions and to address their concerns. As the reviewers point ot in their follow-up comments, the additional discussion regarding experiments on the part of the authors have helped to resolve several of the reviewers' key concerns. The authors are strongly encouraged to ensure that the paper is updated based upon the discussion with the reviewers, which includes updating the paper to include the new experimental results.